# Probabilistic Geometric Principal Component Analysis with application to neural data

**Han-Lin Hsieh**
Ming Hsieh Department of Electrical and Computer Engineering
Viterbi School of Engineering
University of Southern California
Los Angeles, CA, U.S.A
`hanlinhs@usc.edu`

**Maryam M. Shanechi**[*]
Ming Hsieh Department of Electrical and Computer Engineering
Thomas Lord Department of Computer Science
Alfred E. Mann Department of Biomedical Engineering
Viterbi School of Engineering
University of Southern California
Los Angeles, CA, U.S.A
`shanechi@usc.edu`

## Abstract

Dimensionality reduction is critical across various domains of science including neuroscience. Probabilistic Principal Component Analysis (PPCA) is a prominent dimensionality reduction method that provides a probabilistic approach unlike the deterministic approach of PCA and serves as a connection between PCA and Factor Analysis (FA). Despite their power, PPCA and its extensions are mainly based on linear models and can only describe the data in a Euclidean coordinate system around the mean of data. However, in many neuroscience applications, data may be distributed around a nonlinear geometry (i.e., manifold) rather than lying in the Euclidean space around the mean. We develop Probabilistic Geometric Principal Component Analysis (PGPCA) for such datasets as a new dimensionality reduction algorithm that can explicitly incorporate knowledge about a given nonlinear manifold that is first fitted from these data. Further, we show how in addition to the Euclidean coordinate system, a geometric coordinate system can be derived for the manifold to capture the deviations of data from the manifold and noise. We also derive a data-driven EM algorithm for learning the PGPCA model parameters. As such, PGPCA generalizes PPCA to better describe data distributions by incorporating a nonlinear manifold geometry. In simulations and brain data analyses, we show that PGPCA can effectively model the data distribution around various given manifolds and outperforms PPCA for such data. Moreover, PGPCA provides the capability to test whether the new geometric coordinate system better describes the data than the Euclidean one. Finally, PGPCA can perform dimensionality reduction and learn the data distribution both around and on the manifold. These capabilities make PGPCA valuable for enhancing the efficacy of dimensionality reduction for analysis of high-dimensional data that exhibit noise and are distributed around a nonlinear manifold, especially for neural data.

## 1 Introduction

There exist numerous well-established algorithms for dimensionality reduction designed to efficiently identify principal components that explain crucial features in high-dimensional (high-D) data in $\mathbb{R}^n$. Distinct features necessitate different algorithms. Among them, Principal Component Analysis (PCA) (Greenacre et al., 2022) and maximum likelihood Factor Analysis (FA) (Bartholomew et al., 2011) are

---

[*]Corresponding Author.

widely recognized. PCA, grounded in a deterministic model, maximizes a critical feature—the data variance—explained by its principal components. In contrast, FA, rooted in a probabilistic model, efficiently captures another important feature—the correlation between elements in the data—via its loading matrix (analogous to principal components in PCA) by maximizing the data log-likelihood. This fundamental difference results in PCA and FA being employed in deterministic and probabilistic analyses separately.

The gap stemming from the fundamental difference in the model assumptions of PCA and FA is bridged by Probabilistic Principal Component Analysis (PPCA) (Tipping & Bishop, 1999b). PPCA, a special type of FA with a *uniform* diagonal noise covariance, proves that this new condition renders FA's loading matrix equal to PCA's principal components (Tipping & Bishop, 1999b). This insight enhances PCA's value both theoretically and practically. Now, PCA's principal components maximize not only the data variance but also the data log-likelihood given an underlying FA probabilistic model. Moreover, these principal components (equivalent to the loading matrix in PPCA) can be computed analytically, unlike FA's loading matrices, which generally require numerical solutions (De Winter & Dodou, 2012).

All of these methods are widely used in real-world applications including in neuroscience for analyses of neural population activity. However, PCA, FA, and PPCA are all grounded in linear models composed of a Euclidean coordinate system around the mean of data. As such, they do not capture nonlinear structure in data. Further, this linear assumption explains why these algorithms do not require the selection of a coordinate system, as all linear bases of $\mathbb{R}^n$ are equivalent under linear transformations. However, in many applications including in neuroscience, data may be distributed around a nonlinear manifold rather than lying in the Euclidean space around the mean. For example, neural population activity has been shown to be distributed around a ring manifold in the head direction system (Chaudhuri et al., 2019; Jensen et al., 2020) or around a Torus manifold in the hippocampus (Gardner et al., 2022). Indeed, knowledge about the manifold can be provided through various existing methods based on data. For example, the type of manifold underlying noisy data can be identified by the topological data analysis (TDA) (Singh et al., 2008), and that manifold can be fitted by splines (Zheng et al., 2012; He & Shi, 1996; Bojanov et al., 2013) or other graph-based methods (Yin et al., 2008; Fefferman et al., 2018; 2023). However, incorporating such knowledge of a nonlinear manifold that is first fitted from data within the PPCA framework remains challenging to date.

While a few studies have explored nonlinear extensions of PPCA, they have assumed that data must lie *precisely on top* of a specific manifold without deviations from it (Zhang & Fletcher, 2013; Zhang et al., 2019; Nodehi et al., 2020). However, in many applications including in neuroscience, rather than being on top of a manifold, data is distributed around it with some deviation and with noise. Thus, these prior PPCA extensions have not found application for these datasets such as neural activity. To model these datasets, we need to find a way to not only incorporate a given manifold, but also derive a coordinate system – which we term distribution coordinates – to capture the deviation outside of the manifold. Indeed, it may be possible to compute a coordinate systems attached to this nonlinear manifold that is not equivalent to the Euclidean coordinate system and that better describes the data; this is different from the case of the linear model in PPCA in which all coordinate systems are equivalent under linear transformations.

**Contributions** Here we address the above challenges by introducing Probabilistic Geometric Principal Component Analysis (PGPCA). PGPCA generalizes PPCA. Given a nonlinear manifold that is first fitted from data, PGPCA can incorporate this manifold with distribution coordinates that are computed for this manifold in its probabilistic model. PGPCA achieves dimensionality reduction by maximizing the data log-likelihood. Due to the nonlinear manifold, the Singular Value Decomposition (SVD) used in PPCA cannot be used to find the loading matrix in PGPCA. Thus, we derive an Expectation-Maximization (EM) algorithm to compute the PGPCA loading matrix. Further, we show how in addition to the Euclidean distribution coordinate, a geometric distribution coordinate can be derived for the manifold to capture the deviations of data from the manifold and noise. Due to the nonlinear manifold/geometry, the geometric and Euclidean distribution coordinates yield different data log-likelihood values. As such, we show how we can compute this log-likelihood and use it as a metric for distinguishing the distribution coordinates in a data-driven manner.

We structure this paper as follows. In Section 3, we first provide a detailed mathematical derivation of PGPCA, including its probabilistic model and the corresponding EM learning algorithm. In Section

4, we demonstrate the success of PGPCA with simulations of multiple manifolds and analyses on neural population data from the mouse head direction system (Peyrache et al., 2015; Chaudhuri et al., 2019). We also show that PGPCA outperforms the existing PPCA framework by capturing the geometry in both simulations and real data. Finally, we illustrate PGPCA's ability to distinguish between geometric and Euclidean distribution coordinates in simulations and real data. In Section 5, we present a summary and discuss limitations.

## 2 RELATED WORK

Various extensions have been developed based on PPCA. Ahn & Oh (2003) modifies the PPCA EM algorithm to more efficiently compute the PCA principal components in order. To improve the interpretation of PPCA, prior studies have made its loading matrix sparse by, for example, restricting the domain of the probabilistic distribution in the E-step of PPCA EM (Khanna et al., 2015) or adding penalty terms in the cost function (Park et al., 2017). Penalizing the PPCA EM cost function has also been used in finding the efficient PPCA model dimension (Deng & Craiu, 2023). A supervised version of PPCA (Yu et al., 2006) has also been developed for labeled data. Zhang et al. (2017) has focused on using the mixture PPCA (Tipping & Bishop, 1999a) to integrate two monitoring statistics in order to address a fault diagnosis problem. However, all of the above extensions are based on the PPCA linear model lying in the Euclidean space around the mean. As such, these works cannot incorporate the nonlinear manifold underlying the data for dimensionality reduction and modeling, which is what we enable here.

In addition to the above, a few studies have explored extending PPCA to include specific nonlinear manifolds. Probabilistic principal geodesic analysis (PPGA) (Zhang & Fletcher, 2013; Fletcher & Zhang, 2016) extends principal geodesic analysis (PGA) (Fletcher et al., 2003) into a probabilistic framework. Mixture PPGA (Zhang et al., 2019) combines multiple PPGA models. Nodehi et al. (2020) develops the PPCA linear model within the Torus $\mathbb{T}^n$ space, as opposed to the $\mathbb{R}^n$ space, thereby extending torus PCA (Eltzner et al., 2018) to a probabilistic context. However, all these approaches require data to lie precisely on top of a specific manifold without any deviation from it. This assumption is not the case in many applications such as neuroscience, where neural activity data are distributed around manifolds with deviation and also exhibit noise. As such these prior methods have not found application to such datasets such as neural activity. Our method PGPCA is designed for such datasets and models observations that are probabilistically distributed around a given manifold that is first fitted from data. Unlike the above studies, Lawrence & Hyvärinen (2005) develops the Gaussian process latent variable model (GP-LVM), another nonlinear probabilistic model inspired by PPCA. The nonlinearity is encoded by a kernel function between the latent states in GP-LVM. As these latent states are treated as parameters rather than random variables, GP-LVM is typically used for categorization tasks rather than for distribution modeling, which is our goal. Given these disparate assumptions about the distribution of observations relative to the manifold and the properties/roles of the latent states, PGPCA addresses a distinct application and thus serves a complementary role compared with these prior studies.

## 3 METHODOLOGY

We first define the notations and the probabilistic model of PGPCA. Then we derive its log-likelihood and evidence lower bound (ELBO) for the EM algorithm. Finally, we summarize PGPCA EM by providing a pseudo code (Algorithm 1).

### 3.1 PGPCA PROBABILISTIC MODEL

We define the PGPCA model as

$$\boldsymbol{y}_t = \boldsymbol{\phi}(\boldsymbol{z}_t) + \boldsymbol{K}(\boldsymbol{z}_t) \times \boldsymbol{C} \times \boldsymbol{x}_t + \boldsymbol{r}_t \tag{1}$$

where all notations are listed in Table 1. Briefly, we have $T$ observations $\boldsymbol{y}_{1:T} \in \mathbb{R}^n$. Each $\boldsymbol{y}_t$ is composed of three parts. The first part is the $l$-dimensional manifold $\mathcal{M} = \{\boldsymbol{\phi}(\boldsymbol{z}) \,|\, \forall \boldsymbol{z} \in \Omega_z \subset \mathbb{R}^l\}$ where $\boldsymbol{z}_t \sim p(\boldsymbol{z})$ is the manifold state and a random variable (R.V.) in set $\Omega_z$. Essentially, $\boldsymbol{z}_t$ specifies the location on top of the manifold. Conditioned on $\boldsymbol{z}_t$, the second part is a zero-mean normal distribution $\boldsymbol{K}(\boldsymbol{z}_t)\boldsymbol{C}\boldsymbol{x}_t$ where $\boldsymbol{C}$ is the loading matrix and $\boldsymbol{K}(\boldsymbol{z}_t)$ is the coordinate system for the

Table 1: PGPCA model notations

| Notation | Description | Notation | Description |
|---|---|---|---|
| $\boldsymbol{y}_t \in \mathbb{R}^n$ | observation at time $t \in [1,T]$ | $\boldsymbol{\phi}(\boldsymbol{z}_t) \in \mathbb{R}^n$ | a $l$-dim manifold $\subset \mathbb{R}^n$ |
| $\boldsymbol{z}_t \in \Omega_z \subset \mathbb{R}^l$ | an iid random manifold state $\sim p(\boldsymbol{z})$ | $\boldsymbol{K}(\boldsymbol{z}_t) \in \mathbb{R}^{n \times n}$ | distribution coordinate at $\boldsymbol{z}_t$ |
| $\boldsymbol{x}_t \in \mathbb{R}^m$ | an iid normal R.V. $\sim \mathcal{N}(\boldsymbol{0}, \boldsymbol{I}_m)$ | $\boldsymbol{C} \in \mathbb{R}^{n \times m}$ | loading matrix |
| $\boldsymbol{r}_t \in \mathbb{R}^n$ | an iid normal R.V. $\sim \mathcal{N}(\boldsymbol{0}, \sigma^2 \boldsymbol{I}_n)$ | | |

data distribution around the manifold, termed distribution coordinate, with orthonormal property (i.e., $\boldsymbol{K}(\boldsymbol{z}_t)'\boldsymbol{K}(\boldsymbol{z}_t) = \boldsymbol{K}(\boldsymbol{z}_t)\boldsymbol{K}(\boldsymbol{z}_t)' = \boldsymbol{I}_n$, an identity matrix in $\mathbb{R}^{n \times n}$). Thus, $\boldsymbol{C}$ follows the distribution coordinate $\boldsymbol{K}$ and determines the principal directions that cover most of the $\boldsymbol{y}_{1:T}$ distribution. The third part, $\boldsymbol{r}_t$, with its isotropic variance $\sigma^2$, captures any residual in $\boldsymbol{y}_{1:T}$ that is not already covered. We define the dimension of a PGPCA model $m$ as the dimension of $\boldsymbol{x}_t$ or equivalently the rank of the loading matrix $\boldsymbol{C}$ ($0 \leq m \leq n$). When $m = 0$, $\boldsymbol{C}$ is set to 0. Finally, our PGPCA model covers the PPCA model (Tipping & Bishop, 1999b) as a special case by setting $\boldsymbol{\phi}(\boldsymbol{z}_t) = \boldsymbol{0}$ and $\boldsymbol{K}(\boldsymbol{z}_t) = \boldsymbol{I}_n$. In this case, the model (1) reduces to $\boldsymbol{y}_t = \boldsymbol{C}\boldsymbol{x}_t + \boldsymbol{r}_t$ and the linear hyperplanes/subspaces are modeled by $\boldsymbol{C}\boldsymbol{x}_t$, which is the same as in PPCA. Thus, PGPCA is a generalization of PPCA and extends it from the case where data is assumed to lie around the mean of data – which can be considered as the central manifold in PPCA – to the case where data can lie around nonlinear manifolds.

## 3.2 PGPCA EM: E-STEP

We need to learn a PGPCA model (1) that describes the data the best. We formalize this learning problem as follows: given data $\boldsymbol{y}_{1:T}$, the manifold function $\boldsymbol{\phi}$ (that is first fitted from data), and the distribution coordinate function $\boldsymbol{K}$ (either Euclidean or geometric as we derive later in section 4), find the model parameters $\boldsymbol{C}$, $\sigma^2$, and $p(\boldsymbol{z})$ in (1) by maximizing the data log-likelihood $\mathcal{L} = \ln p(\boldsymbol{y}_{1:T})$. Since $\boldsymbol{y}_t$'s for different $t$'s are iid ($t = 1:T$), we can write the log-likelihood as

$$\mathcal{L} = \sum_{i=1}^{T} \ln p(\boldsymbol{y}_i) = \sum_{i=1}^{T} \ln \int_{\Omega_z} p(\boldsymbol{y}_i|\boldsymbol{z})p(\boldsymbol{z}) \, d\boldsymbol{z} \tag{2}$$

where $p(\boldsymbol{y}_i|\boldsymbol{z})$ is a normal distribution from (1) such that

$$p(\boldsymbol{y}_i|\boldsymbol{z}) = \mathcal{N}(\boldsymbol{\phi}(\boldsymbol{z}), \boldsymbol{\Psi}(\boldsymbol{z})) = \frac{1}{(2\pi)^{\frac{n}{2}}|\boldsymbol{\Psi}(\boldsymbol{z})|^{\frac{1}{2}}} \times e^{-\frac{1}{2}(\boldsymbol{y}_i-\boldsymbol{\phi}(\boldsymbol{z}))'\boldsymbol{\Psi}(\boldsymbol{z})^{-1}(\boldsymbol{y}_i-\boldsymbol{\phi}(\boldsymbol{z}))} \tag{3}$$

$$\boldsymbol{\Psi}(\boldsymbol{z}) = \boldsymbol{K}(\boldsymbol{z})\boldsymbol{C}\boldsymbol{C}'\boldsymbol{K}(\boldsymbol{z})' + \sigma^2 \boldsymbol{I}_n \tag{4}$$

where $'$ indicates the matrix transpose operation. To find the maximum-likelihood parameter estimates, we need to partial differentiate $\mathcal{L}$ w.r.t. model parameters to maximize it; but this differentiation is tricky because the integration in (2) is inside the $\ln$ function. We address this challenge by deriving the ELBO $\mathcal{L}^E$ of $\mathcal{L}$ following the standard EM procedure as

$$\mathcal{L} = \sum_{i=1}^{T} \ln \int_{\Omega_z} q_i(\boldsymbol{z}) \times \frac{p(\boldsymbol{y}_i|\boldsymbol{z})p(\boldsymbol{z})}{q_i(\boldsymbol{z})} \, d\boldsymbol{z}$$

$$\geq \sum_{i=1}^{T} \int_{\Omega_z} q_i(\boldsymbol{z}) \Big[ \ln \big( p(\boldsymbol{y}_i|\boldsymbol{z})p(\boldsymbol{z}) \big) - \ln q_i(\boldsymbol{z}) \Big] d\boldsymbol{z} := \mathcal{L}^E \tag{5}$$

where $q_i(\boldsymbol{z})$ is any probability distribution on $\Omega_z$. From the standard EM procedure (Beal, 2003; McLachlan & Krishnan, 2007), we know $\mathcal{L} = \mathcal{L}^E$ if and only if $q_i(\boldsymbol{z}) = p(\boldsymbol{z}|\boldsymbol{y}_i)$ for $\forall i \in [1,T]$. Therefore, given the model parameters $\boldsymbol{C}$, $\sigma^2$, and $p(\boldsymbol{z})$, the E-step of PGPCA EM is derived as

$$q_i(\boldsymbol{z}) = p(\boldsymbol{z}|\boldsymbol{y}_i) = \frac{p(\boldsymbol{y}_i|\boldsymbol{z}) \, p(\boldsymbol{z})}{\int_{\Omega_z} p(\boldsymbol{y}_i|\boldsymbol{z}) \, p(\boldsymbol{z}) \, d\boldsymbol{z}} \tag{6}$$

$$= \begin{cases} \frac{p(\boldsymbol{y}_i|\boldsymbol{z}_s) \, \omega_s}{\sum_{j=1}^{M} p(\boldsymbol{y}_i|\boldsymbol{z}_j) \, \omega_j} & \text{if } \boldsymbol{z} = \boldsymbol{z}_s \in \{\boldsymbol{z}_{1:M}\} \\ 0 & \text{otherwise} \end{cases} \tag{7}$$

Note that equation (7) follows after discretizing $p(\boldsymbol{z})$, which is provided later in (9). This discretization is necessary in fitting $p(\boldsymbol{z})$ in the M-step and for numerical computations as detailed next.

---

**Algorithm 1** PGPCA EM

---

**Input:** $\boldsymbol{y}_{1:T}$, model dimension $m$, landmark $\boldsymbol{z}_{1:M}$, manifold $\boldsymbol{\phi}(\cdot)$, distribution coordinate $\boldsymbol{K}(\cdot)$.
**Output:** probability $\omega_{1:M}$, parameters $\boldsymbol{C}$ and $\sigma^2$.

---

Initialize $\omega_{1:M}$, $\boldsymbol{C}$, and $\sigma^2$ randomly.
**repeat**
   {E-step}
   Compute $q_i(\boldsymbol{z}_j)$ by (7) for $\forall i \in [1, T]$ & $\forall j \in [1, M]$.
   {M-step}
   Compute $\omega_j$ by (11) for $\forall j \in [1, M]$.
   Compute $\boldsymbol{\Gamma}(q)$ by (14) and then $\mathrm{eig}(\boldsymbol{\Gamma}(q)) = \{\overline{\gamma}_{1:n}\}$ in descending order.
   Compute $\sigma^2$ by (16) and then $\boldsymbol{C}$ by (15).
**until** ELBO $\mathcal{L}^E$ in (5) converges.

---

## 3.3 PGPCA EM: M-STEP TO FIND $p(\boldsymbol{z})$

Given $q_{1:T}(\boldsymbol{z})$ from the E-step, the M-step finds the optimal model parameters $\boldsymbol{C}$ and $\sigma^2$ in addition to $p(\boldsymbol{z})$ to maximize the ELBO $\mathcal{L}^E$. Only the first part of equation (5), $q_i(\boldsymbol{z}) \ln \left(p(\boldsymbol{y}_i|\boldsymbol{z})p(\boldsymbol{z})\right)$, relates to these parameters, so we define

$$\mathcal{L}^M := \sum_{i=1}^{T} \int_{\Omega_z} q_i(\boldsymbol{z}) \ln \left(p(\boldsymbol{y}_i|\boldsymbol{z})p(\boldsymbol{z})\right) d\boldsymbol{z} = \sum_{i=1}^{T} \int_{\Omega_z} q_i(\boldsymbol{z}) \ln p(\boldsymbol{y}_i|\boldsymbol{z}) + q_i(\boldsymbol{z}) \ln p(\boldsymbol{z}) \, d\boldsymbol{z} \quad (8)$$

Parameters $\boldsymbol{C}$ and $\sigma^2$ are only in the first term in (8), which is defined as $\mathcal{L}_1^M$ in (12), and the distribution $p(\boldsymbol{z})$ is only in the second term in (8), which is defined as $\mathcal{L}_2^M$ in (10), respectively. But a challenge here is that we must first parameterize $p(\boldsymbol{z})$ to learn it. To do so, we select $M$ landmarks $\{\boldsymbol{z}_{1:M}\} \subset \Omega_z$ with nonnegative weights $\omega_{1:M}$ such that $\sum_{j=1}^{M} \omega_j = 1$, and discretize $p(\boldsymbol{z})$ as

$$p(\boldsymbol{z}) \approx \sum_{j=1}^{M} \omega_j \times \delta(\boldsymbol{z} - \boldsymbol{z}_j) \quad (9)$$

where $\delta : \mathbb{R}^l \to \{0, 1\}$ is the Dirac delta function. This is how we discretize $q_i(\boldsymbol{z})$ in E-step by substituting (9) into (6) to get (7). So the new M-step goal is: given $q_{1:T}(\boldsymbol{z})$, find parameters $\boldsymbol{C}$, $\sigma^2$, and $\omega_{1:M}$ to maximize $\mathcal{L}^M$. To find the optimized $\omega_{1:M}$, we define $\mathcal{L}_2^M$, the second term in (8), as

$$\mathcal{L}_2^M := \sum_{i=1}^{T} \int_{\Omega_z} q_i(\boldsymbol{z}) \ln p(\boldsymbol{z}) \, d\boldsymbol{z} = \sum_{i=1}^{T} \sum_{j=1}^{M} q_i(\boldsymbol{z}_j) \ln \omega_j \quad (10)$$

Using Lagrange multipliers (Bertsekas, 2014), the optimal $\omega_j$ to maximize $\mathcal{L}_2^M$ is found as

$$\omega_j = \frac{1}{T} \sum_{i=1}^{T} q_i(\boldsymbol{z}_j) \quad \text{for} \quad \forall j \in [1, M] \quad (11)$$

## 3.4 PGPCA EM: COMPUTING THE FIRST TERM IN $\mathcal{L}^M$ TO DERIVE THE M-STEP FOR $\boldsymbol{C}$ AND $\sigma^2$

Next, we solve for parameters $\boldsymbol{C}$ and $\sigma^2$ that maximize $\mathcal{L}_1^M$, the first term of $\mathcal{L}^M$ in (8). Due to the nonlinear manifold and the distribution coordinate $\boldsymbol{K}(\boldsymbol{z})$, finding the model parameters is more challenging than PPCA, which assumes a linear model. We first derive a formula for $\mathcal{L}_1^M$ (c.f. (13)), and then optimize it to find $\boldsymbol{C}$ and $\sigma^2$ in the next section. First, we expand $\mathcal{L}_1^M$ using (3) as follows:

$$\mathcal{L}_1^M := \sum_{i=1}^{T} \int_{\Omega_z} q_i(\boldsymbol{z}) \ln p(\boldsymbol{y}_i|\boldsymbol{z}) \, d\boldsymbol{z}$$

$$= -\frac{1}{2} \times \sum_{i=1}^{T} \int_{\Omega_z} q_i(\boldsymbol{z}) \times \left[ n \ln 2\pi + \ln |\boldsymbol{\Psi}_z| + (\boldsymbol{y}_i - \boldsymbol{\phi}_z)' \boldsymbol{\Psi}_z^{-1} (\boldsymbol{y}_i - \boldsymbol{\phi}_z) \right] d\boldsymbol{z} \quad (12)$$

where we use the simplified notations $\phi(z) \equiv \phi_z$ from (3) and $\Psi(z) \equiv \Psi_z$ from (4) for ease of exposition. The right-hand side consists of three parts that are added together. We compute these three parts of (12) one by one in appendix A. From there, we have

$$\mathcal{L}_1^M = -\frac{T}{2} \times \left\{ n \ln 2\pi + (n-m) \ln \sigma^2 + \ln |\underbrace{\sigma^2 \boldsymbol{I}_m + \boldsymbol{C}'\boldsymbol{C}}_{\text{define as } \boldsymbol{\Omega}}| + \mathrm{tr}\left[ \underbrace{(\sigma^2 \boldsymbol{I}_n + \boldsymbol{C}\boldsymbol{C}')^{-1}}_{\text{define as } \boldsymbol{\Lambda}} \times \boldsymbol{\Gamma}(q) \right] \right\} \quad (13)$$

$$\boldsymbol{\Gamma}(q) = \frac{1}{T} \sum_{i=1}^{T} \sum_{j=1}^{M} \boldsymbol{\Gamma}_{i,z_j} \times q_i(\boldsymbol{z}_j) \quad \text{where} \quad \boldsymbol{\Gamma}_{i,z} = \boldsymbol{K}_z'(\boldsymbol{y}_i - \boldsymbol{\phi}_z)(\boldsymbol{y}_i - \boldsymbol{\phi}_z)'\boldsymbol{K}_z \quad (14)$$

Critically, compared to (12) where we started from, our derivations (appendix A) lead to all summations and integrations being captured in $\boldsymbol{\Gamma}(q)$, which is interestingly independent of parameters $\boldsymbol{C}$ and $\sigma^2$. This derivation makes partial differentiating $\mathcal{L}_1^M$ w.r.t. $\boldsymbol{C}$ and $\sigma^2$ much easier and tractable, thus solving the major M-step challenge for learning the model parameters in the general case that includes nonlinear manifolds. Moreover, formula (13) is the same as PPCA log-likelihood (Tipping & Bishop, 1999a), except for the matrix $\boldsymbol{\Gamma}(q)$. This makes solving for $\boldsymbol{C}$ and $\sigma^2$ easy. We show this in detail in section 3.5.

## 3.5 PGPCA EM: M-STEP FOR $\boldsymbol{C}$ AND $\sigma^2$

Now we are ready to find the optimal $\boldsymbol{C}$ and $\sigma^2$ by maximizing $\mathcal{L}_1^M$ in (13). Critically, our derivation showed that we can summarize all the nonlinear manifold and distribution coordinate information in one term $\boldsymbol{\Gamma}(q)$ within the $\mathcal{L}_1^M$. As such, interestingly, (13) becomes a generalization of the PPCA log-likelihood in Tipping & Bishop (1999a) in that they have the same formula except that our $\boldsymbol{\Gamma}(q)$ considers the manifold and the distribution coordinate on it, while PPCA's matrix $\boldsymbol{S}$ in Tipping & Bishop (1999a) does not. Therefore, we can solve for our optimal $\boldsymbol{C}$ and $\sigma^2$ using the PPCA formula, and all the established guarantees in the PPCA theory also apply to this nonlinear manifold case. Here we write the optimal solution of $\boldsymbol{C}$ and $\sigma^2$ directly. The detailed derivation is in appendix B.

Define $\overline{\gamma}_{1:n}$ as the eigenvalues of $\mathrm{eig}(\boldsymbol{\Gamma}(q))$ in descending order. The optimal $\boldsymbol{C}$ is derived as

$$\boldsymbol{C} = \boldsymbol{U}\boldsymbol{D} \quad \text{where} \quad \begin{cases} \boldsymbol{\Gamma}(q)\,\boldsymbol{u}_i = \overline{\gamma}_i\,\boldsymbol{u}_i \\ d_i = \sqrt{\overline{\gamma}_i - \sigma^2} \end{cases} \quad \forall i \in [1, m] \quad (15)$$

where $\boldsymbol{D} = \mathrm{diag}(d_{1:m})$ and $\boldsymbol{u}_i$ is the $i^{\text{th}}$ column of $\boldsymbol{U}$ and the $i^{\text{th}}$ eigenvector of $\boldsymbol{\Gamma}(q)$. The optimal $\sigma^2$ is

$$\sigma^2 = \frac{1}{n - m} \times \sum_{i=m+1}^{n} \overline{\gamma}_i \quad (16)$$

Our pseudo code summarizes PGPCA EM in Algorithm 1. The intuitive explanation behind our solution is that the loading matrix $\boldsymbol{C}$ captures the dominant directions in data $\boldsymbol{y}_{1:T}$ distribution around the manifold $\phi$, and $\sigma^2$ captures the residual directions with their average variance as a noise term. Since all steps in Algorithm 1 are analytical, PGPCA EM is efficient in terms of training time (appendix C), similar to classical EM for linear state-space models (Roweis & Ghahramani, 1999).

## 4 EXPERIMENTS

We show that our PGPCA model plus its EM algorithm can solve four problems: (1) Given data $\boldsymbol{y}_{1:T}$, PGPCA EM can learn an $m$-dimensional probabilistic model that includes a given underlying nonlinear manifold $\phi$ and a distribution coordinate $\boldsymbol{K}(z)$, which can be either Euclidean or computed according to our geometric distribution coordinate. (2) It allows us to perform hypothesis testing to select the Euclidean or geometric distribution coordinate by fitting alternative PGPCA models with two different $\boldsymbol{K}(z)$'s, and selecting the one with the higher data log-likelihood $\mathcal{L}$ in (2). (3) We can perform dimensionality reduction by fitting a low-dimensional PGPCA model with any dimension $m \in [0, n]$ that is as low as the user desires. (4) PGPCA EM can not only learn the data distribution around the manifold but also the distribution on the manifold; indeed, we show that the weights of

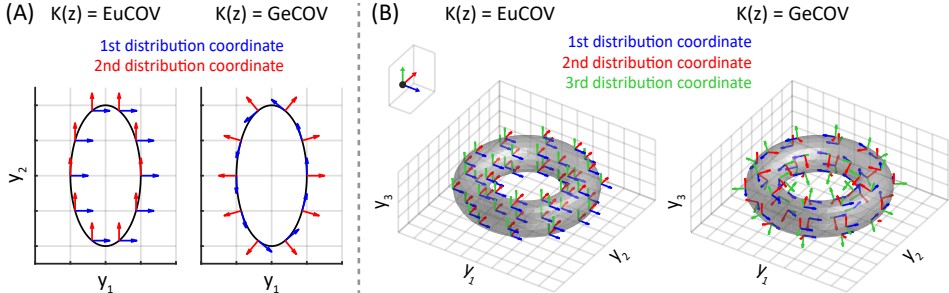

Figure 1: Distribution coordinate $\boldsymbol{K}(\boldsymbol{z})$ can be Euclidean (EuCOV) or geometric (GeCOV) on a loop and a torus. (A) When $\boldsymbol{K}(z) = \text{EuCOV}$ on a loop $\subset \mathbb{R}^2$, it always aligns with the embedding coordinate $\mathbb{R}^2$ no matter where it is on the loop. In contrast, if $\boldsymbol{K}(z) = \text{GeCOV}$, the distribution coordinate follows the tangent vector and the normal vector. (B) Again, $\boldsymbol{K}(\boldsymbol{z})$ can always align to the axes of $\mathbb{R}^3$ (EuCOV) or be composed of two tangent vectors plus another vector perpendicular to the torus surface (GeCOV). The top-left inset figure shows the PPCA case whose manifold is the mean of data (the black dot) with its only distribution coordinate system, which is equal to EuCOV.

Table 2: PGPCA (GeCOV/EuCOV), PPCA, and FA log-likelihood of full-rank models ($m = n$)

| True $\Big\{$ | loop in $\mathbb{R}^2$ | | loop in $\mathbb{R}^{10}$ | | torus in $\mathbb{R}^3$ | | data analysis | |
|---|---|---|---|---|---|---|---|---|
| | GeCOV | EuCOV | GeCOV | EuCOV | GeCOV | EuCOV | Mouse12 | Mouse28 |
| GeCOV | **-2.931** | -2.725 | **-27.921** | -28.484 | **-5.626** | -5.560 | **-31.758** | **-24.752** |
| EuCOV | -2.939 | **-2.698** | -27.993 | **-28.356** | -5.631 | **-5.523** | -31.908 | -25.089 |
| PPCA | -3.048 | -2.991 | -31.945 | -31.677 | -5.862 | -5.907 | -34.622 | -29.316 |
| FA | -3.048 | -2.991 | -31.945 | -31.677 | -5.862 | -5.907 | -34.615 | -29.310 |

manifold latent state distribution $\omega_{1:M}$ (from discretizing $p(\boldsymbol{z})$ in (9)) can be jointly learned with parameters $\boldsymbol{C}$ and $\sigma^2$ in (1) and result in a similar log-likelihood as the true model.

We show that PGPCA can solve the above four problems using neural data analyses and extensive simulations covering various nonlinear manifolds, distribution coordinates, and manifold latent state distributions $p(\boldsymbol{z})$. The nonlinear manifolds include a loop (in $\mathbb{R}^2$ or $\mathbb{R}^{10}$) and a torus. The distribution coordinate $\boldsymbol{K}(z)$ can be Euclidean (EuCOV) or geometric (GeCOV) (see Figure 1 and appendix C). For the torus, its $p(\boldsymbol{z})$ has two options: a uniform distribution on the angular space $[0, 2\pi] \times [0, 2\pi]$ (uniAng), or a uniform distribution on the torus surface (uniTorus) (Figure 5 in the appendix). The real dataset includes neural spike firing rates recorded from anterodorsal thalamic nucleus (ADn) of mice, a part of the thalamo-cortical head-direction (HD) circuit, while animals were exploring an open environment (Peyrache et al., 2015; Peyrache & Buzsáki, 2015). The firing rates are projected to $\mathbb{R}^{10}$ following the same preprocessing as that in prior work (Chaudhuri et al., 2019). Details of neural data analyses and simulations are in appendix C. A summary of the log-likelihoods for each model, based on the neural data analyses and simulations, is presented in Table 2. In this table, all models are set to full rank ($m = n$) to maximize their log-likelihoods, which makes PPCA mathematically equivalent to FA. However, slight differences in the log-likelihoods between PPCA and FA are observed since their models are learned numerically.

## 4.1 PGPCA FINDS THE CORRECT DISTRIBUTION COORDINATES ON A 1D LOOP

We first show that PGPCA EM can learn a nonlinear probabilistic model from data and distinguish different distribution coordinates in hypothesis testing. To show that our method succeeds in incorporating the manifold, we compare with the widely used PPCA, which is linear. Figure 2A shows the 2D probability distribution from the true models and from the learned models by PGPCA/PPCA. First, we see that the learned PGPCA model's distributions are closer to the true model's distribution compared to PPCA's distribution, no matter what the distribution coordinates (GeCOV/EuCOV) in the true or learned PGPCA models are. This demonstrates the importance of modeling data probabilistically with an underlying nonlinear manifold as enabled by PGPCA. Moreover, the true model's

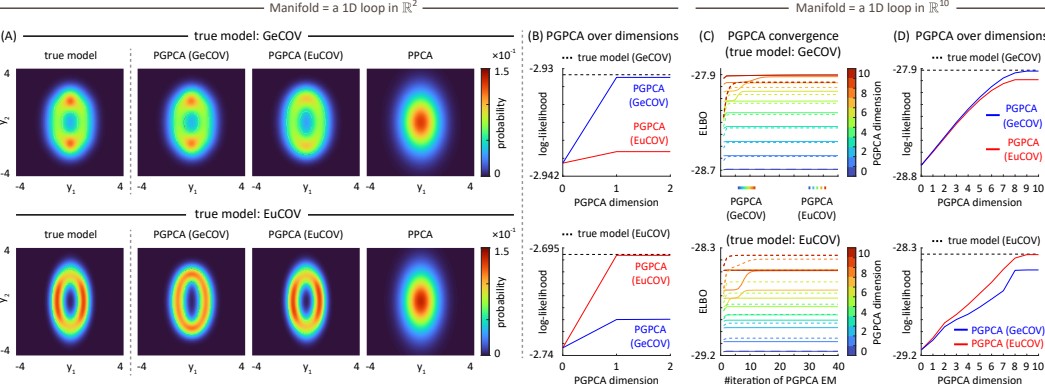

Figure 2: PGPCA can recover the true model distribution, distinguish different distribution coordinates $\boldsymbol{K}(z)$, and perform dimensionality reduction simultaneously. Across all panels (A–D), the true model's $\boldsymbol{K}(z)$ are GeCOV and EuCOV in the top and the bottom row, respectively. (A) The PGPCA (GeCOV) model learned by EM recovers the true model distribution while PPCA does not. Further, PGPCA can do so only with the correct $\boldsymbol{K}(z)$, showing that PGPCA can distinguish the correct coordinate. (B) PGPCA with the correct $\boldsymbol{K}(z)$ always has higher trial-average log-likelihood (paired t-test: top and bottom $< 1.7 \times 10^{-12}$). (C) Learned PGPCA models with different dimensions $m \in [0, 10]$ (color bar) and with different $\boldsymbol{K}(z)$ (EuCOV/GeCOV) converge within 40 EM iterations. (D) The same conclusion in (B) also holds here when the loop $\subset \mathbb{R}^{10}$ (paired t-test: top and bottom $< 3.1 \times 10^{-4}$).

distribution is only recovered by the learned PGPCA model when their distribution coordinates match. Figure 2B and Table 2 further confirm that the learned PGPCA model with the correct distribution coordinate $\boldsymbol{K}(z)$ has higher log-likelihood than the learned PGPCA model with the incorrect one. As such, fitting the two alternative PGPCA models and comparing their log-likelihood can successfully distinguish the true distribution coordinate underlying the data. This shows PGPCA's ability to solve problems (1) and (2) listed at the beginning of section 4.

## 4.2 PGPCA CAN PERFORM DIMENSIONALITY REDUCTION.

For dimensionality reduction, we simulate a true model built on a loop embedded in $\mathbb{R}^{10}$, so we have enough dimensions for this application. Figure 2C shows that regardless of whether the true model is GeCOV or EuCOV, PGPCA EM can converge under any PGPCA model dimension $m \in [0, 10]$. Thus, this EM method can robustly learn a PGPCA model with any dimension. In Figure 2D, the learned PGPCA model with the correct $\boldsymbol{K}(z)$ always has a higher log-likelihood compared to the alternative, even when the PGPCA dimension $m$ is selected to be low. Therefore, PGPCA can still distinguish the correct distribution coordinate $\boldsymbol{K}(z)$ even when its dimension is chosen low. This result shows that PGPCA can simultaneously perform both dimensionality reduction and distribution coordinate selection, solving problem (3) stated at the beginning of section 4.

## 4.3 PGPCA CAN RECOVER THE TRUE MODEL'S DISTRIBUTION EVEN WHILE LEARNING $p(\boldsymbol{z})$.

Figure 3A shows that PGPCA EM can recover the true model's distribution when its $\boldsymbol{K}(\boldsymbol{z})$ matches the true one, regardless of whether $p(\boldsymbol{z})$ is given or learned. We also find that PGPCA can again distinguish the correct coordinate system $\boldsymbol{K}(\boldsymbol{z})$ even when $p(\boldsymbol{z})$ is being jointly learned. This shows that PGPCA EM can learn not only the distribution around the manifold, but also the distribution $p(\boldsymbol{z})$ on the manifold. This solves problem (4) stated at the beginning of section 4.

Furthermore, Figure 3B shows that the learned PGPCA model with its $\boldsymbol{K}(\boldsymbol{z})$ matched to the true one always has a higher log-likelihood than the learned PGPCA model with the unmatched $\boldsymbol{K}(\boldsymbol{z})$, whether $p(\boldsymbol{z})$ is learned or not. Thus, the hypothesis testing ability of PGPCA EM in distinguishing different distribution coordinates also holds even when simultaneously learning $p(\boldsymbol{z})$. The average performance across uniAng/uniTorus and given/learned $p(\boldsymbol{z})$ is provided in Table 2.

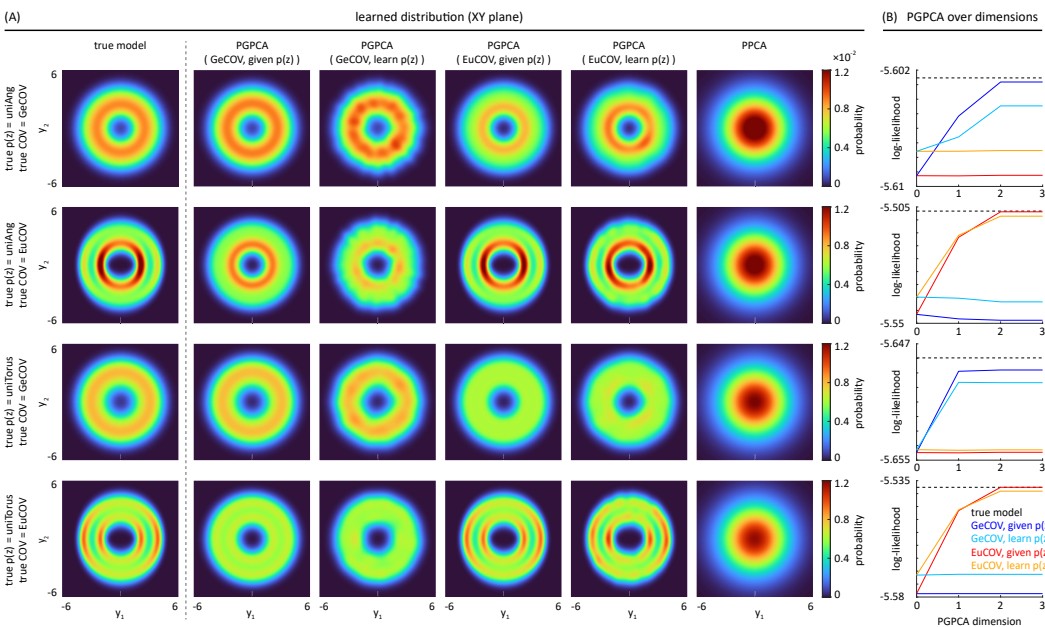

Figure 3: PGPCA EM can recover the true model's distribution even when simultaneously learning the manifold state probability $p(z)$, while PPCA does not. (A) The first row shows that when the true model's $p(z) =$ uniAng and $K(z) =$ GeCOV, the learned PGPCA model's distribution is similar to the true one, regardless of whether $p(z)$ is given (column 2) or learned (column 3). Also, this is the case only if PGPCA's $K(z)$ is GeCOV (true coordinate), showing its ability to identify the true coordinate. Rows 2–4 show the same conclusion for alternative true models having different $p(z)$ and $K(z)$. (B) The trial-average log-likelihood of the four learned PGPCA models (columns 2–5 in (A)). Again, the learned PGPCA model whose $K(z)$ matches the true one always has higher log-likelihood than the unmatched PGPCA model, regardless of whether $p(z)$ is given or learned, showing hypothesis testing capability. For all 4 rows with given or learned $p(z)$, paired t-test $< 2.4 \times 10^{-7}$.

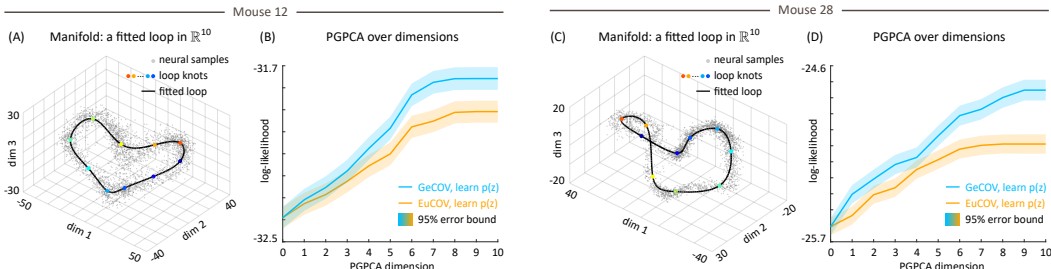

Figure 4: PGPCA (GeCOV) better captures the distribution of neural firing rates in mice head direction circuit. (A) The fitted loop manifold in $\mathbb{R}^{10}$ and the neural data distributed around it. (B) PGPCA (GeCOV) model consistently has higher log-likelihood than PGPCA (EuCOV) model across all dimensions. (C) and (D) are the same as (A) and (B) for a second mouse, with conclusions being the same.

## 4.4 PGPCA CAN DISTINGUISH DISTRIBUTION COORDINATES ON REAL DATA

We applied PGPCA to neural firing rates recorded from the thalamo-cortical head direction circuit of six mice, and we select two mice as examples here. First, we found that the main manifold structure was a loop, consistent with prior work, and so fitted this loop using a cubic spline with 10 knots selected by K-means (appendix C). Figures 4A and 4C display the projected neural firing rates along with the fitted manifolds, which are the 1D loops embedded in $\mathbb{R}^{10}$. The neural firing rates are distributed not precisely on, but around, the manifold, indicating that the main manifold alone

is insufficient for completely modeling noisy data. This observation underscores the necessity of PGPCA, which captures the deviation outside of the manifold through distribution coordinates and noise. So we constructed the distribution coordinate and ran PGPCA EM, and compared with PPCA and FA.

Table 2 and Figure 6 in the appendix demonstrate that PGPCA significantly outperforms PPCA and FA. Further and interestingly, Figures 4B and 4D show that PGPCA GeCOV more accurately captures the firing rates than PGPCA EuCOV across both mice. This latter result suggests that the noise not accounted for by the main loop manifold also originates from the same geometric structure rather than being in the Euclidean space. This inference can only be made using a model with a coordinate system around the main manifold, which is a major capability provided by PGPCA. This again shows that PGPCA can also perform hypothesis testing about the coordinate system in which data is distributed. These conclusions again held on the other mice (Table 3 in the appendix).

## 5 CONCLUSION

We developed PGPCA, a method that generalizes the widely-used PPCA for analyses of data that are distributed around a given nonlinear manifold that is fitted from data. Unlike PPCA, which assumes that data lies around the mean in Euclidean space, PGPCA incorporates the nonlinear manifold as well as distribution coordinates attached to this manifold to capture deviations from it and noise. Also, in addition to being able to use the Euclidean coordinate around the manifold, PGPCA can also compute a geometric coordinate system around the manifold, which we derived here. Finally, PGPCA can perform hypothesis testing to pick between the Euclidean and the geometric distribution coordinates based on which can better describe the data distribution. In this paper, we focused on the Euclidean (EuCOV) and our geometric (GeCOV) $K(z)$ because they naturally arise from the linear embedding space $\mathbb{R}^n$ and the underlying nonlinear manifold, respectively. If prior knowledge about the data suggests the hypothesis that another new form of $K(z)$ is needed (assuming it can be derived), PGPCA can serve as a tool to validate or reject this hypothesis by comparing its log-likelihood with that of other $K(z)$ options, such as EuCOV and GeCOV. Our PGPCA can accommodate new $K(z)$ because in deriving the PGPCA EM algorithm, we did not impose any specific assumptions on $K(z)$ beyond the basic orthonormal property.

We demonstrated the success of PGPCA and its efficient analytical EM learning algorithm on real neural firing rate data and over three types of simulated manifolds with different manifold state distribution $p(z)$ (uniAng/uniTorus) and different distribution coordinates $K(z)$ (EuCOV/GeCOV). Our results show that PGPCA can correctly i) fit the nonlinear probabilistic model, ii) distinguish between Euclidean and geometric distribution coordinates, iii) perform dimensionality reduction, and iv) learn the manifold state distribution on top and around the manifold. Further, in both simulations and real neural data, PGPCA outperformed PPCA by capturing the manifold. One major application of PGPCA is for modeling of neural data time-series in the fields of neuroscience and neurotechnology given the evidence that neural data distribute around nonlinear manifolds (section 1). However, PGPCA is not limited to neural data time-series and can in principle be applied to any time-series dataset with a data-fitted underlying manifold. A limitation of PGPCA, similarly to PPCA and PCA, is that it is a static dimensionality reduction method and thus does not explicitly model the auto-correlations in data. Further, similar to these methods, PGPCA assumes that the data distribution is stable over time. Further work can extend PGPCA to enable manifold-based dynamical analyses (Abbaspourazad et al., 2024; Sani et al., 2024) or adaptive modeling to track non-stationarity in the data distribution (Ahmadipour et al., 2021; Yang et al., 2021; Degenhart et al., 2020). Finally, PGPCA allows for incorporation of manifold knowledge, which should first be obtained based on data using existing manifold identification and fitting methods (e.g., TDA and splines). As we show in our real neural data analyses, incorporating this knowledge can substantially improve dimensionality reduction and distribution modeling.

### ACKNOWLEDGMENTS

This work was partly supported by the National Institutes of Health (NIH) grants DP2-MH126378 and RF1DA056402, and by the Army Research Office (ARO) under contract W911NF-16-1-0368 as part of the collaboration between the US DOD, the UK MOD and the UK Engineering and Physical Research Council (EPSRC) under the Multidisciplinary University Research Initiative (MURI).

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

# A  DERIVE THE CONCISE FORM OF $\mathcal{L}_1^M$

In this appendix, we compute three parts of (12) one by one below to transform (12) into (13).

**The first part $\mathcal{L}_{1,1}^M$.** Since $q_i(\boldsymbol{z})$ is a probability distribution on $\Omega_z$ and $n \ln 2\pi$ is a constant, we have

$$\mathcal{L}_{1,1}^M := \sum_{i=1}^{T} \int_{\Omega_z} q_i(\boldsymbol{z}) \times n \ln 2\pi \, d\boldsymbol{z} = T \times n \ln 2\pi \tag{17}$$

**The second part $\mathcal{L}_{1,2}^M$.** Recall that for the coordinate system, $\boldsymbol{K}_z' \boldsymbol{K}_z = \boldsymbol{I}_n$. Following (4) and the matrix determinant lemma (Harville, 1998), we have

$$
\begin{aligned}
|\boldsymbol{\Psi}_z| &= |\boldsymbol{K}_z \boldsymbol{C} \boldsymbol{C}' \boldsymbol{K}_z' + \sigma^2 \boldsymbol{I}_n| \\
&= |\sigma^2 \boldsymbol{I}_n| \times |\boldsymbol{I}_m + \boldsymbol{C}' \boldsymbol{K}_z' \times (\sigma^2 \boldsymbol{I}_n)^{-1} \times \boldsymbol{K}_z \boldsymbol{C}| \\
&= (\sigma^2)^n \times |(\sigma^2)^{-1} \times (\sigma^2 \boldsymbol{I}_m + \boldsymbol{C}' \boldsymbol{C})| \\
&= (\sigma^2)^{n-m} \times |\sigma^2 \boldsymbol{I}_m + \boldsymbol{C}' \boldsymbol{C}|
\end{aligned}
\tag{18}
$$

Interestingly, this derivation shows that $|\boldsymbol{\Psi}_z|$ is a constant independent of $\boldsymbol{z}$. Therefore, the second part simplifies to

$$\mathcal{L}_{1,2}^M = T \times \left[ (n-m) \ln \sigma^2 + \ln |\sigma^2 \boldsymbol{I}_m + \boldsymbol{C}' \boldsymbol{C}| \right] \tag{19}$$

**The third part $\mathcal{L}_{1,3}^M$.** The key idea here is to rewrite the vector norm weighted by $\boldsymbol{\Psi}_z^{-1}$, i.e.,

$$\|\boldsymbol{y}_i - \boldsymbol{\phi}_z\|_{\boldsymbol{\Psi}_z^{-1}} := (\boldsymbol{y}_i - \boldsymbol{\phi}_z)' \boldsymbol{\Psi}_z^{-1} (\boldsymbol{y}_i - \boldsymbol{\phi}_z),$$

using the identity $\mathrm{tr}(\boldsymbol{AB}) = \mathrm{tr}(\boldsymbol{BA})$ (when $\boldsymbol{AB}$ and $\boldsymbol{BA}$ are well-defined) (Petersen et al., 2008), which gives the following

$$\|\boldsymbol{y}_i - \boldsymbol{\phi}_z\|_{\boldsymbol{\Psi}_z^{-1}} := (\boldsymbol{y}_i - \boldsymbol{\phi}_z)' \boldsymbol{\Psi}_z^{-1} (\boldsymbol{y}_i - \boldsymbol{\phi}_z) = \mathrm{tr}\Big[ \boldsymbol{\Psi}_z^{-1} \underbrace{(\boldsymbol{y}_i - \boldsymbol{\phi}_z)(\boldsymbol{y}_i - \boldsymbol{\phi}_z)'}_{\text{define as } \boldsymbol{\Pi}_{i,z}} \Big] \tag{20}$$

Remember $\boldsymbol{K}_z \boldsymbol{K}_z' = \boldsymbol{I}_n$, so the inverse of $\boldsymbol{\Psi}_z$ in (4) is

$$\boldsymbol{\Psi}_z^{-1} = (\boldsymbol{K}_z \boldsymbol{C} \boldsymbol{C}' \boldsymbol{K}_z' + \sigma^2 \boldsymbol{K}_z \boldsymbol{K}_z')^{-1} = \boldsymbol{K}_z \times (\sigma^2 \boldsymbol{I}_n + \boldsymbol{C} \boldsymbol{C}')^{-1} \times \boldsymbol{K}_z' \tag{21}$$

Now we transform (20) into

$$\mathrm{tr}\Big[ \boldsymbol{\Psi}_z^{-1} \boldsymbol{\Pi}_{i,z} \Big] = \mathrm{tr}\Big[ \boldsymbol{K}_z (\sigma^2 \boldsymbol{I}_n + \boldsymbol{C} \boldsymbol{C}')^{-1} \boldsymbol{K}_z' \boldsymbol{\Pi}_{i,z} \Big] = \mathrm{tr}\Big[ (\sigma^2 \boldsymbol{I}_n + \boldsymbol{C} \boldsymbol{C}')^{-1} \underbrace{\boldsymbol{K}_z' \boldsymbol{\Pi}_{i,z} \boldsymbol{K}_z}_{\text{define as } \boldsymbol{\Gamma}_{i,z}} \Big] \tag{22}$$

From (20) and (22), since the trace, summation, and integral operators are linear and can be swapped, the third part $\mathcal{L}_{1,3}^M$ can be written as

$$\mathcal{L}_{1,3}^M = \sum_{i=1}^{T} \int_{\Omega_z} q_i(\boldsymbol{z}) \, \mathrm{tr}\Big[ \boldsymbol{\Psi}_z^{-1} \boldsymbol{\Pi}_{i,z} \Big] d\boldsymbol{z} = T \times \mathrm{tr}\Big[ (\sigma^2 \boldsymbol{I}_n + \boldsymbol{C} \boldsymbol{C}')^{-1} \times \boldsymbol{\Gamma}(q) \Big] \tag{23}$$

where,

$$\boldsymbol{\Gamma}(q) = \frac{1}{T} \sum_{i=1}^{T} \int_{\Omega_z} \boldsymbol{\Gamma}_{i,z} \, q_i(\boldsymbol{z}) \, d\boldsymbol{z} \tag{24}$$

Finally, since $\mathcal{L}_1^M = -\frac{1}{2} \times \left( \mathcal{L}_{1,1}^M + \mathcal{L}_{1,2}^M + \mathcal{L}_{1,3}^M \right)$, $\mathcal{L}_1^M$ in (12) is equal (13) by combining the derived forms above for the three parts in (17), (19), and (23).

The last thing to notice is that $q_i(\boldsymbol{z})$ is discretized in (7) for numerical computations. So in practice, we can compute $\boldsymbol{\Gamma}(q)$ defined in (24) numerically as (14). This completes the derivation.

## B    DERIVE OPTIMAL $C$ AND $\sigma^2$ IN PGPCA MODEL

In this appendix, we derive the optimal $C$ and $\sigma^2$ by maximizing $\mathcal{L}_1^M$ in (13). Our (13) and the PPCA log-likelihood in Tipping & Bishop (1999a) have the same formula except that our $\Gamma(q)$ considers the manifold and the distribution coordinate on it, while PPCA's matrix $S$ in Tipping & Bishop (1999a) does not. Therefore, we can solve for our optimal $C$ and $\sigma^2$ using the PPCA formula. We rewrite the derivation for completeness and notation consistency below. For more details, please refer to Tipping & Bishop (1999a).

We first optimize $C$. Using matrix calculus operations, we have

$$\frac{\partial \mathcal{L}_1^M}{\partial C} = -\frac{T}{2} \times \left[ 2\Lambda^{-1}C - 2\Lambda^{-1}\Gamma(q)\Lambda^{-1}C \right] = \mathbf{0} \tag{25}$$

So the optimal $C$ satisfies the following condition

$$\Gamma(q)\Lambda^{-1}C = C \tag{26}$$

We define the SVD of $C = UDV'$ where $U \in \mathbb{R}^{n \times m}$, $D \in \mathbb{R}^{m \times m}$ is diagonal, and $V \in \mathbb{R}^{m \times m}$. By Woodbury matrix identity (Petersen et al., 2008), we have

$$\begin{aligned}
\Lambda^{-1}C &= (\sigma^2 I_n + CC')^{-1}C \\
&= C \times (\sigma^2 I_m + C'C)^{-1} \\
&= UDV' \times \left[ V(\sigma^2 I_m + D^2)V' \right]^{-1} \\
&= UD \times (\sigma^2 I_m + D^2)^{-1}V'
\end{aligned} \tag{27}$$

Substituting (27) in (26) and multiplying $V(\sigma^2 I_m + D^2)D^{-1}$ on both sides, we have

$$\begin{aligned}
\Gamma(q)U &= C \times V(\sigma^2 I_m + D^2)D^{-1} \\
&= UD \times (\sigma^2 I_m + D^2)D^{-1} \\
&= U \times (\sigma^2 I_m + D^2)
\end{aligned} \tag{28}$$

Note that $D$ and $\sigma^2 I_m + D^2$ can be swapped because they are diagonal. From (28) and $C = UDV'$, we conclude that

1. **$V$ can be any orthonormal matrix in $\mathbb{R}^{m \times m}$.** For convenience, we set $V = I_m$.
2. **Columns of $U$ are eigenvectors of $\Gamma(q)$.** Define eigenvalues $\mathrm{eig}(\Gamma(q)) = \{\gamma_{1:n}\}$ *without order (e.g., ascending/descending)*. Then $U = [u_1 | \ldots | u_m]$ ($u_i$ is the i$^{\text{th}}$ column of $U$) and $D = \mathrm{diag}(d_{1:m})$ such that $u_i$ is the eigenvector w.r.t. eigenvalue $\gamma_i = \sigma^2 + d_i^2$ from (28)

In summary, given $\sigma^2$, the optimal $C$ is

$$C = UD \quad \text{where} \quad \begin{cases} \Gamma(q)\,u_i = \gamma_i\,u_i \\ d_i = \sqrt{\gamma_i - \sigma^2} \end{cases} \quad \forall i \in [1, m] \tag{29}$$

where $u_i$, $\gamma_i$, and $d_i$ are defined above.

The next step is optimizing $\sigma^2$. To do so, we substitute $C$ from (29) into (13), and then rewrite $\mathcal{L}_1^M$ as

$$\mathcal{L}_1^M = -\frac{T}{2} \times \left\{ n \ln 2\pi + (n - m) \ln \sigma^2 + \sum_{i=1}^{m} \ln \gamma_i + \frac{1}{\sigma^2} \times \sum_{i=m+1}^{n} \gamma_i + m \right\} \tag{30}$$

Setting $\frac{\partial \mathcal{L}_1^M}{\partial (\sigma^2)} = 0$, the optimal $\sigma^2$ is

$$\sigma^2 = \frac{1}{n - m} \times \sum_{i=m+1}^{n} \gamma_i \tag{31}$$

The remaining challenge now is that the optimal $C$ and $\sigma^2$ in (29) and (31) do not complete the answer yet because we also have to select $\gamma_{1:m}$ from $\mathrm{eig}(\Gamma(q))$. The power of our derivation for $\mathcal{L}_1^M$

formula in (13) is that because the manifold and the distribution coordinate are summarized in the $\mathbf{\Gamma}(q)$ term, we can use the results in Tipping & Bishop (1999a) directly to select $\gamma_{1:m}$. Briefly, to do this $\gamma_{1:m}$ selection, we substitute (31) into (30) to rewrite $\mathcal{L}_1^M$ again as

$$\mathcal{L}_1^M = -\frac{T}{2} \times \left\{ n \ln 2\pi + \sum_{i=1}^n \ln \gamma_i - \sum_{i=m+1}^n \ln \gamma_i + n + (n-m) \times \ln \left( \frac{1}{n-m} \times \sum_{i=m+1}^n \gamma_i \right) \right\}$$

(32)

Note that $\sum_{i=1}^n \ln \gamma_i$ is a constant because $\{\gamma_{1:n}\} = \mathrm{eig}(\mathbf{\Gamma}(q))$. Therefore, maximizing $\mathcal{L}_1^M$ in (32) is equivalent to minimizing

$$\ln \left( \frac{1}{n-m} \times \sum_{i=m+1}^n \gamma_i \right) - \frac{1}{n-m} \sum_{i=m+1}^n \ln \gamma_i$$

(33)

which is a Jensen's inequality (Needham, 1993; Jensen, 1906). It's proved in Tipping & Bishop (1999a) that the optimal $\gamma_{m+1:n}$ for (33) must be a consecutive series in $\mathrm{eig}(\mathbf{\Gamma}(q))$. More precisely, defining $\overline{\gamma}_{1:n}$ as the descending series of $\mathrm{eig}(\mathbf{\Gamma}(q))$, we have that $\gamma_{m+1:n}$ is a consecutive series in $\overline{\gamma}_{1:n}$.

Finally, from (29) and (31), we see that

$$\forall j \in [1, m], \ \gamma_j \geq \sigma^2 = \frac{1}{n-m} \times \sum_{i=m+1}^n \gamma_i$$

(34)

Therefore, $\gamma_{1:m}$ cannot include $\overline{\gamma}_n$, the smallest eigenvalue of $\mathbf{\Gamma}(q)$, so $\overline{\gamma}_n \in \gamma_{m+1:n}$. Combined with the consecutive condition on $\gamma_{m+1:n}$, we can conclude that $\gamma_i = \overline{\gamma}_i$ for $\forall i = [1, n]$. Then (29) and (31) become (15) and (16), respectively. This completes the M-step of PGPCA EM.

## C    SETTING OF ALL SIMULATION CASES AND DATA ANALYSIS.

We describe the details of simulations and data analysis below. For simulations, we simulate 3 kinds of manifolds: a 1D loop in $\mathbb{R}^2$, a 1D loop in $\mathbb{R}^{10}$, and a 2D torus in $\mathbb{R}^3$. First, we rewrite the PGPCA model (1) as

$$\boldsymbol{y}_t = \boldsymbol{\phi}(\boldsymbol{z}_t) + \boldsymbol{K}(\boldsymbol{z}_t) \times (\boldsymbol{C}\boldsymbol{x}_t + \boldsymbol{r}_t)$$

(35)

because $\boldsymbol{K}(\boldsymbol{z}_t) \times \sigma^2 \boldsymbol{I}_n \times \boldsymbol{K}(\boldsymbol{z}_t)' = \sigma^2 \boldsymbol{I}_n$, so the covariance of $\boldsymbol{K}(\boldsymbol{z}_t) \, \boldsymbol{r}_t$ and $\boldsymbol{r}_t$ are the same. Note that

$$\mathrm{Cov}(\boldsymbol{C}\boldsymbol{x}_t + \boldsymbol{r}_t) = \boldsymbol{C}\boldsymbol{C}' + \sigma^2 \boldsymbol{I}_n = \mathbf{\Lambda}$$

(36)

Therefore, every simulation case is specified by the manifold function $\boldsymbol{\phi}$ with the manifold latent state distribution $p(\boldsymbol{z})$ on top of it, distribution coordinate $\boldsymbol{K}(\boldsymbol{z})$, and basic covariance $\mathbf{\Lambda}$. We describe all simulation cases for the above three manifolds below.

**A 1D loop embedded in $\mathbb{R}^2$.** We define $z \in [0, 2\pi]$ with $p(z) = U(0, 2\pi)$ where $U(a, b)$ is a continuous uniform distribution within $[a, b]$. The nonlinear manifold is an ellipse with function $\phi(z) = [\cos(z), 2\sin(z)]$. The basic covariance is taken as $\mathbf{\Lambda} = \mathrm{diag}([0.1, 0.3])$. The distribution coordinate $\boldsymbol{K}(z)$ for the simulated data can be Euclidean or geometric. Euclidean means $\boldsymbol{K}(z) = \boldsymbol{I}_2$, and so the distribution coordinate system follows the embedded Euclidean coordinate at all points $z$ on the manifold. We refer to this scenario as the Euclidean covariance (EuCOV) since $\mathbf{\Lambda}$ follows the Euclidean coordinate (Figure 1A, *left*). On the contrary, the geometric case refers to when $\boldsymbol{K}(z)$ is composed of the tangent and normal vectors at each $z$ along the manifold, which is an ellipse here. We refer to this alternative scenario as the geometric covariance (GeCOV, Figure 1A, *right*). For convenience, we also say that $\boldsymbol{K}(z)$ is EuCOV or GeCOV when it's the Euclidean or geometric coordinate, respectively. Because there are two options for $\boldsymbol{K}(z)$ corresponding to the EuCOV and GeCOV scenarios respectively, we will fit two types of PGPCA models with PGPCA using either a Euclidean or a geometric $\boldsymbol{K}(z)$, which we term PGPCA EuCOV and PGPCA GeCOV, respectively; this thus leads to two simulation cases for this ellipse in $\mathbb{R}^2$. For both cases, we generate 5000 training samples from the true model and learn every PGPCA model (EuCOV/GeCOV) with 500 landmarks $z_{1:500}$ in (9) using 20 EM iterations. The PGPCA model dimension can be $m \in [0, 2]$.

**A 1D loop embedded in $\mathbb{R}^{10}$.** To simulate this loop in higher dimensional space, we define the manifold points $z \in [0, L]$ with $p(z) = U(0, L)$ where $L$ is the length of the loop. We form the manifold $\phi(z)$ as a cubic spline with 6 knots, and with length $L$. The basic covariance is $\mathbf{\Lambda} = \mathrm{diag}([20, 2, 18, 4, \ldots, 12, 10])$. The PGPCA model can be either EuCOV or GeCOV. In this case, the $\boldsymbol{K}(z)$ in a GeCOV model is computed using the Gram-Schmidt process (Leon et al., 2013) with the tangent vector as the first vector, and the Euclidean axes $\boldsymbol{e}_{1:10}$ as the other independent vectors to be orthogonalized by the Gram-Schmidt process one by one in sequence. Again, there are two simulation cases w.r.t. this spline in $\mathbb{R}^{10}$ corresponding to EuCOV or GeCOV being the true distribution coordinate, respectively. For both cases, we generate 5000 samples from the true model, and learn every PGPCA model (EuCOV/GeCOV) with 500 landmarks $z_{1:500}$ using 40 EM iterations. The dimension of PGPCA model can be $m \in [0, 10]$.

**A 2D torus embedded in $\mathbb{R}^3$.** Defining $\boldsymbol{z} \in [0, 2\pi] \times [0, 2\pi]$, the torus manifold is given by Do Carmo (2016)

$$\boldsymbol{\phi}(\boldsymbol{z}) = [(3 + \cos z_2) \cos z_1, (3 + \cos z_2) \sin z_1, \sin z_2]$$

Here the basic covariance $\mathbf{\Lambda} = \mathrm{diag}([0.1, 0.3, 0.5])$ and $\boldsymbol{K}(z)$ can be either EuCOV or GeCOV (Figure 1B). GeCOV $\boldsymbol{K}(z)$ is composed of two tangent vectors ($\frac{\partial \phi}{\partial z_1}$ and $\frac{\partial \phi}{\partial z_2}$) and their cross product. We also give $p(\boldsymbol{z})$ two options: a uniform distribution on the angular space $[0, 2\pi] \times [0, 2\pi]$ (uniAng), or a uniform distribution on the torus surface (uniTorus). Because we have two options for $\boldsymbol{K}(\boldsymbol{z})$ and $p(\boldsymbol{z})$, there are $2 \times 2 = 4$ simulation cases w.r.t. this torus in $\mathbb{R}^3$. Figure 5 shows the true model distributions under the 4 cases. For all four cases, we generate 50000 samples from the true model, and every PGPCA model (uniAng/uniTorus $\times$ EuCOV/GeCOV) with 1000 landmarks $z_{1:1000}$ is learned using 40 EM iterations. We increase the number of training samples because we need to fit $\omega_{1:1000} = p(\boldsymbol{z}_{1:1000})$ in all four cases. The dimension of PGPCA model can be $m \in [0, 3]$.

**Performance measures in simulations.** After learning the PGPCA model from the training samples using the PGPCA EM algorithm with one of three manifolds above, for each simulation case, we generate 20 test trials from the true model. Each trial includes 2000 samples. For each of the 20 trials, we measure the performance of a learned PGPCA model with the average log-likelihood defined as $\mathcal{L}/T$ with $T = 2000$ being the trial length. In the figures, all log-likelihoods for the learned PGPCA models are the average of these 20 trial-average log-likelihoods, and comparisons are done with paired t-tests between the trial-average log-likelihood groups from two different learned PGPCA models (i.e., 20 trials in the paired t-test comparisons).

For data analysis, we utilized the neural firing rates recorded from mice's brains. This dataset is publicly available (Peyrache & Buzsáki, 2015), and further details can be found in Peyrache et al. (2015). The preprocessing steps prior to applying PGPCA are primarily based on Chaudhuri et al. (2019). These steps are summarized below for completeness.

**Data.** For all 6 mice, spikes were recorded from intracortical shanks implanted in the anterodorsal thalamic nucleus (ADn), a part of the thalamo-cortical head-direction (HD) circuit, while the mice were exploring an open environment. There are 8 shanks with 50 cells for Mouse 12 and 4 shanks with 22 cells for Mouse 28. The sampling rate is 20 kHz. The numbers of shanks and cells of other mice are listed in Table 3. All mouse data have the same preprocessing and PGPCA training and testing procedures.

**Preprocessing.** We followed the preprocessing steps in prior work. For both mice, we first computed the firing rates by smoothing the spike time-series with a Gaussian kernel with a standard deviation of 100 ms. The firing rates were then down-sampled to 15000 samples with a 100 ms step size (equivalent to data-duration of 25 minutes in total). As preprocessing following prior work, we first applied a square root on the firing rates to stabilize the variance (Chaudhuri et al., 2019), and then projected the data using Isomap (Tenenbaum et al., 2000) from $\mathbb{R}^{50}$ (Mouse 12) and $\mathbb{R}^{22}$ (Mouse 28) to $\mathbb{R}^{10}$. This 10D space is the space in which PGPCA operates, as shown in Figure 4.

**PGPCA training and testing.** We split the 15000 samples equally into 5 trials for 5-fold cross-validation. In each fold, we concatenated 4 trials to form a training set. Similar to what has been observed previously (Chaudhuri et al., 2019), we found that neural data was distributed around a loop manifold, but had both noise and deviations from it. We thus fitted a 1D loop in $\mathbb{R}^{10}$ using K-means (Hastie et al., 2009) with 10 clusters. The means of these clusters served as the knots of a closed cubic spline. We determined the order for connecting these knots by solving the traveling salesman problem (Hahsler & Hornik, 2007). This resulted in a manifold model $\phi$ constructed by

Table 3: PGPCA (GeCOV/EuCOV) and PPCA log-likelihood of full-rank models ($m = n$)

| True { | data analysis | | | | | |
|---|---|---|---|---|---|---|
|  | Mouse12 | Mouse17 | Mouse20 | Mouse24 | Mouse25 | Mouse28 |
| shanks | 8 | 8 | 8 | 4 | 4 | 4 |
| cells | 50 | 29 | 9 | 10 | 10 | 22 |
| GeCOV | **-31.758** | **-30.407** | **-19.751** | **-18.687** | **-20.298** | **-24.752** |
| EuCOV | -31.908 | -30.595 | -19.768 | -18.702 | -20.356 | -25.089 |
| PPCA | -34.622 | -32.668 | -21.560 | -21.134 | -24.977 | -29.316 |

Table 4: PGPCA computational complexity for every iteration

|  | **computational step** | **time complexity** |
|---|---|---|
| E-step | Compute $q_i(z_j)$ for $\forall i \in [1, T]$ and $\forall j \in [1, M]$. | $\mathcal{O}(TM^2)$ |
| M-step | Compute $\omega_j$ for $\forall j \in [1, M]$. | $\mathcal{O}(TM)$ |
|  | Compute $\mathbf{\Gamma}(q)$. | $\mathcal{O}(TMn^2)$ |
|  | Compute $\mathrm{eig}(\mathbf{\Gamma}(q))$. | $\mathcal{O}(n^3)$ (Parlett, 2000) |
|  | Compute $\sigma^2$ and $\mathbf{C}$. | $\mathcal{O}(n - m)$ and $\mathcal{O}(nm)$ |

a closed cubic spline. We built the distribution coordinates (EuCOV/GeCOV) in the same manner as in the simulation of a loop in $\mathbb{R}^{10}$ (appendix C). We then trained the model using the PGPCA EM algorithm. After model training, in each cross validation fold, we assessed performance using the 3000 samples in the test set. As our performance measure, we used the data log-likelihood. The average performance was computed as the average of log-likelihoods over the 15000 samples across the 5 test sets in the 5 cross-validation folds. We compared PGPCA models with two distinct distribution coordinates, Euclidean (EuCOV) and Geometric (GeCOV). We also compared with PPCA. Comparisons between two different learned PGPCA or PPCA models were conducted using paired t-tests on the log-likelihoods of the 15000 test samples.

**PGPCA training time.** It takes about 13 minutes on a regular desktop computer to learn a 10D PGPCA (GeCOV) model with 12000 training samples. This shows that because all steps in Algorithm 1 are analytical, PGPCA EM is efficient in terms of training time, similar to classical EM for linear state-space models (Roweis & Ghahramani, 1999). The theoretical computational complexity of each step in one PGPCA EM iteration (Algorithm 1) is listed in Table 4, and every PGPCA EM iteration's computational complexity is $\mathcal{O}(TM^2) + \mathcal{O}(TMn^2)$. Note that PPCA is not iterative (unlike EM), and its computational complexity is $\mathcal{O}(Tn^2)$. Additionally, because the E-step of our PGPCA EM can find the posterior distribution $q_i(\mathbf{z}_j)$ and our M-step can analytically find the parameters that maximize the ELBO, together our E-step and M-step ensure a monotonic increase in ELBO $\mathcal{L}^E$ with each iteration. Consequently, the sequence of $\mathcal{L}^E$ values is monotonically increasing and bounded by $\mathcal{L}$ from above, ensuring PGPCA EM convergence by the completeness property of real numbers (Rudin et al., 1964).

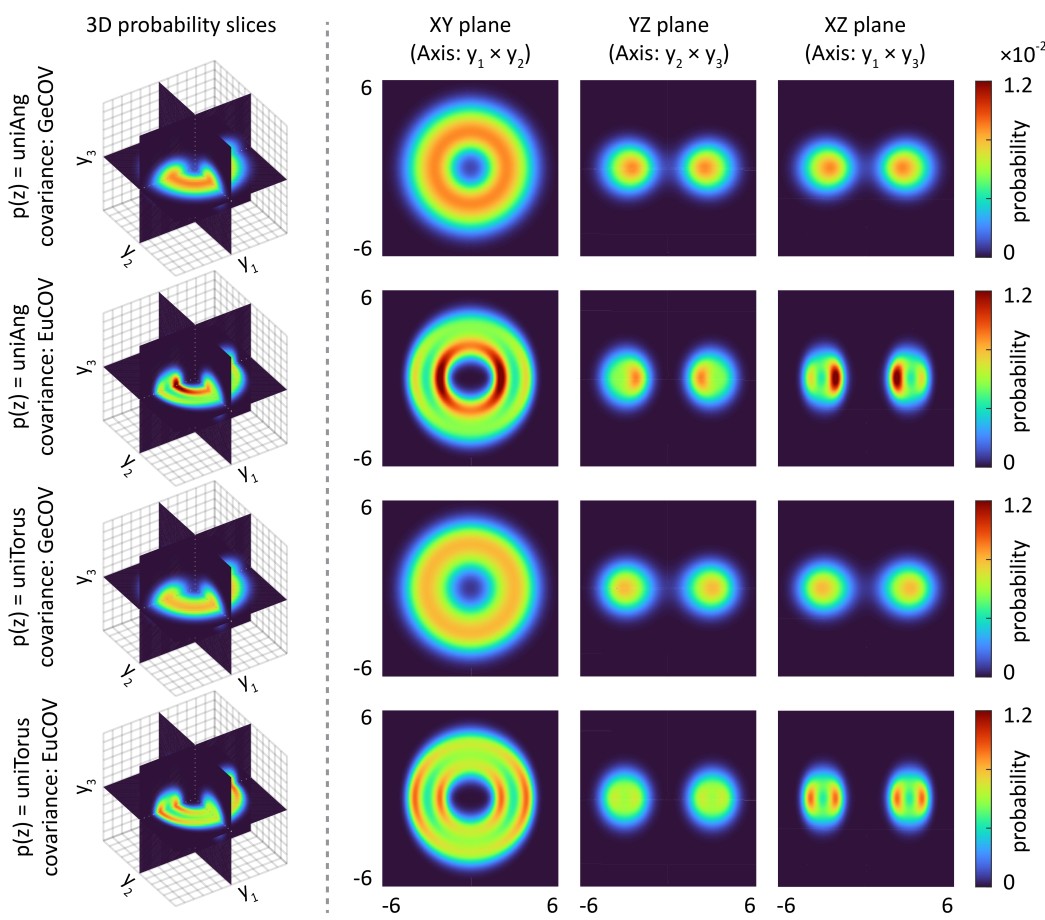

Figure 5: Probability distribution $p(\boldsymbol{y})$ of true models under various manifold latent state probability distributions $p(\boldsymbol{z})$ and various distribution coordinates $\boldsymbol{K}(\boldsymbol{z})$. For each model, we show three slices (XY, YZ, and XZ planes) that go through the 3D probability distribution for visualization. From the XY plane, it's clear that EuCOV makes $p(\boldsymbol{y})$ more directional along the Y axis, and GeCOV is more cylindrically symmetric. Similarly, $p(\boldsymbol{z}) = $ uniAng makes $p(\boldsymbol{y})$ much denser in the inner ring compared to the outer ring, while $p(\boldsymbol{z}) = $ uniTorus does not.

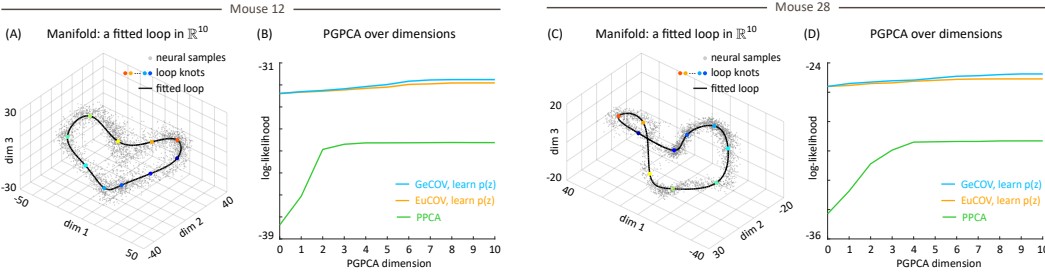

Figure 6: PGPCA (GeCOV and EuCOV) better capture the distribution of neural firing rates in mice head direction circuit compared with PPCA. (A) and (C) are the same fitted loop manifolds as in Figure 4A and 4C. (B) PGPCA models consistently has much higher log-likelihood than PPCA across all dimensions. (D) is the same as (B) for a second mouse, with conclusions being the same.

