# OpenReview forum: "Probabilistic Geometric Principal Component Analysis with application to neural data"
_ICLR.cc/2025/Conference — ICLR 2025 Spotlight_

### Official Review · Reviewer_DfNu · 2024-10-25

**Soundness:** 3
**Presentation:** 3
**Contribution:** 3
**Rating:** 8
**Confidence:** 3

**Summary:**

This paper proposes a generalization of probabilistic principal component analysis (PPCA) to data that lies on a (nonlinear) manifold rather than in Euclidean space around the mean.

The key contributions of the paper are (i) the introduction of the PGPCA model, (ii) an expectation-maximization algorithm to fit the proposed model and (iii) empirical experiments with real and simulated data to compare PGPCA with vanilla PPCA.

**Strengths:**

- The problem of dimensionality reduction for data on a manifold is well motivated, and the proposed method closes the gap between PPCA and data on manifolds.
- The proposed extension PPCA is innovative and well explained.
- The mathematical derivation of the EM-algorithm is concise and understandable, and the empirical experiments show promising results.

**Weaknesses:**

- The simulated data seems to be favorable for PGPCA compared to PPCA as it lies on non-linear manifolds. It remains open how well PGPCA compares to PPCA for data in other settings, e.g. in linear subspaces.
- PGPCA is only compared to PPCA and not other - potentially stronger - baseline methods. Especially, comparisons with other dimension reduction methods for data on manifolds is missing.
- The evaluation on real data is rather limited. Only brain signals of two mice were used.

**Questions:**

- Please add literature on dimensionality reduction for data on a manifold.
- How well does PGPCA work with other choices of $K(z)$ than geometric and Euclidean?
- What theoretical guarantees for the hypothesis test on EuCOV vs. GeCOV are there? (e.g. in terms of statistical power/or asymptotic level)
- How does PGPCA compare to PPCA for data in linear subspaces?
- How does PGPCA compare against stronger competitors?
- How well does PGPCA work on other real data?
- Can the i.i.d. assumptions in model (1) be relaxed? E.g. by allowing (temporal) dependence or slightly varying distributions?
- Can the assumption of normality be relaxed to allow for non-normal errors?

---

> ### Author Response · Authors · 2024-11-22
>
> We thank the reviewer for considering our manuscript and for their feedback. We respond to the reviewer comments below:
>
> &nbsp;
> ## Weaknesses
> &nbsp;
>
> 1. The simulated data seems to be favorable for PGPCA compared to PPCA as it lies on non-linear manifolds. It remains open how well PGPCA compares to PPCA for data in other settings, e.g. in linear subspaces.
>
> We thank the reviewer for raising this point, which is important to clarify. Our PGPCA model encompasses the PPCA model as a special case by setting $\phi(z_t) = \mathbf{0}$ and $K(z_t) = \mathbf{I}_n$. This is explained in Section 3.1 (lines 158–161). Briefly, when the underlying manifold of the data is a linear subspace (i.e., hyperplane), it can be directly modeled as $Cx_t$ with $K(z_t) = \mathbf{I}_n$ in model (1). In other words, the manifold $\phi(z_t)$ captures the nonlinear component in the data $y_t$, while $K(z_t) \times Cx_t$ linearly accounts for the deviation from $\phi(z_t)$. Therefore, in the linear subspace case (hyperplane case), PGPCA performs equally as well as PPCA. This is why we did not include this scenario in our simulations. We will clarify this point about the linear subspace case in Section 3.1.
>
> ---
> 2. PGPCA is only compared to PPCA and not other - potentially stronger - baseline methods. Especially, comparisons with other dimension reduction methods for data on manifolds is missing.
>
> We thank the reviewer for this suggestion. As proposed by reviewer VSUw, we applied the Gaussian Process Latent Variable Model (GP-LVM) (Lawrence, 2005), a nonlinear probabilistic method inspired by PPCA, to our mouse data. The results show that the log-likelihood achieved by GP-LVM was consistently much smaller (less than $-200$) compared to that of our PGPCA (greater than $-35$). This is because for an $n$-dimensional observation $y_t$, GP-LVM models the time series of each observation dimension $y_{1:T}(d) \phantom{s} (d = 1 \dots n)$ separately and in $\mathbb{R}^T$, rather than modeling the distribution of $y_t$ in the embedding space $\mathbb{R}^n$. As a result, it does not efficiently capture the manifold underlying the observations $y_{1:T}$ as our PGPCA does. Note that this is expected, as the goal of GP-LVM is to find a nonlinear low-dimensional description of data for classification, rather than to describe the data distribution in its own embedding space. Indeed, as shown in Lawrence (2005), the log-likelihood of a trained GP-LVM can be very negative around $-500$. As such, our PGPCA outperforms GP-LVM for modeling of data distribution, and thus provides a tool with a distinct goal compared with GP-LVM.
>
> Furthermore, GP-LVM's training time poses a significant bottleneck due to its non-convex cost function and the large number of latent states requiring optimization. For instance, on a regular desktop computer, training a 4D GP-LVM model (i.e., with a latent state dimension of 4) with 3000 time-samples ($T=3000$) takes 532 minutes (approximately 8.9 hours) to converge, as it involves fitting $4 \times 3000 = 12000$ latent state elements. This is because GP-LVM treats its latent states as parameters in the optimization process. In contrast, training our 10D PGPCA (GeCOV) model with 12000 samples takes only about 13 minutes to converge. That is, with more training samples, our PGPCA still converges much faster than GP-LVM. This highlights that PGPCA is better suited for modeling data distributions by leveraging data-fitted manifold information. Indeed, GP-LVM is typically used for categorization tasks rather than for distribution modeling.
>
> ---
> 3. The evaluation on real data is rather limited. Only brain signals of two mice were used.
>
> We thank the reviewer for this comment. We would like to make two points.
>
> First, PGPCA is motivated by the extensive evidence in neuroscience that suggests brain signal time-series data (i.e., neural population activity) often distributes around nonlinear manifolds (see also introduction lines 59-68). As such, we focused on applying PGPCA to brain signals in this paper. We believe PGPCA will be of broad utility in the fields of neuroscience, neuroAI, and neurotechnology.
>
> Second, following the reviewer comment, we have now significantly expanded our real-data evaluations by including 4 additional brain datasets from 4 additional different animals. Table below shows the log-likelihood values for these datasets, similar to our original Table 2 for the original 2 datasets. We see that our conclusions again hold in these new datasets.
>
> | |Mouse17|Mouse20|Mouse24|Mouse25|
> |-|:-:|:-:|:-:|:-:|
> |GeCOV|$\mathbf{-30.407}$|$\mathbf{-19.751}$|$\mathbf{-18.687}$|$\mathbf{-20.298}$|
> |EuCOV|$-30.595$|$-19.768$|$-18.702$|$-20.356$|
> |PPCA|$-32.668$|$-21.560$|$-21.134$|$-24.977$|

---

> ### Author Response · Authors · 2024-11-22
>
> ## Questions
> &nbsp;
>
> 1. Please add literature on dimensionality reduction for data on a manifold.
>
> We appreciate the reviewer’s suggestion. Since PGPCA is a probabilistic model designed to efficiently model data distributions by incorporating data-fitted manifolds, the literature included in the Related Work section focuses on dimensionality reduction methods based on probabilistic models. As mentioned above, we now compare with GP-LVM and include its citation there as well. To our knowledge, we have cited the distinct methods that relate to probabilistic PCA and include nonlinearity and/or manifolds, including mixture PPCA (Tipping & Bishop, 1999), PPCA with penalty terms (Park et al., 2017), torus PPCA (Nodehi et al., 2020), and mixture PPGA (Zhang & Fletcher, 2013). We are happy to cite any other references that the reviewer feels are relevant.
>
> ---
> 2. How well does PGPCA work with other choices of $K(z)$ than geometric and Euclidean?
>
> This is another excellent question. First, we clarify that the geometric $K(z)$ is a new distribution coordinate system that we derived here, and so we compared it with the Euclidean distribution coordinate system as a baseline. Because the Euclidean coordinate already exists, it is currently the alternative distribution coordinate that we could use in PGPCA, in addition to our own new geometric coordinate. Second, if one were to derive another new distribution coordinate system $K(z)$ in the future, PGPCA can also accommodate that in its formulation because, in deriving the PGPCA EM algorithm, we did not impose any specific assumptions on $K(z)$ beyond the basic orthonormal property ($K(z)’K(z) = \mathbf{I}_n$). Third, in this manuscript, we focused on the Euclidean (EuCOV) and our geometric (GeCOV) $K(z)$ because they naturally arise from the linear embedding space $\mathbb{R}^n$ and the underlying nonlinear manifold, respectively. If prior knowledge about the data suggests the hypothesis that another new form of $K(z)$ is needed (assuming it can be derived), PGPCA can serve as a tool to validate or reject this hypothesis by comparing its log-likelihood with that of other $K(z)$ options, such as EuCOV and GeCOV. We will highlight these points in the Conclusion section.
>
> ---
> 3. What theoretical guarantees for the hypothesis test on EuCOV vs. GeCOV are there? (e.g. in terms of statistical power/or asymptotic level)?
>
> This is a good question. As demonstrated in our simulations, regardless of whether the ground-truth model’s $K(z)$ is EuCOV or GeCOV, PGPCA with the correct $K(z)$ achieves higher log-likelihoods when the samples are generated from their respective EuCOV or GeCOV models. This indicates that the PGPCA model does not favor any specific distribution coordinate, making it unbiased for hypothesis testing. Additionally, asymptotically, the log of the likelihood-ratio test converges to the Kullback-Leibler (KL) divergence, which is zero if and only if the modeling distribution matches the true data distribution. In other words, asymptotically, when there is a large number of samples, the true log-likelihood is the upper bound of the modeling log-likelihood, and the latter will asymptotically converge to the former if and only if the modeling distribution is true. Therefore, if the true distribution coordinate is EuCOV, PGPCA will not mistakenly identify GeCOV as the correct $K(z)$, and vice versa.
>
> ---
> 4. How does PGPCA compare to PPCA for data in linear subspaces?
>
> We answer this question in the weakness point 1. Please refer to it.
>
> ---
> 5. How does PGPCA compare against stronger competitors?
>
> We answer this question in the weakness point 2. Please refer to it.
>
> ---
> 6. How well does PGPCA work on other real data?
>
> We answer this question in the weakness point 3. Please refer to it.

---

> ### Author Response · Authors · 2024-11-22
>
> 7. Can the i.i.d. assumptions in model (1) be relaxed? E.g. by allowing (temporal) dependence or slightly varying distributions?
>
> We thank the reviewer for this important question. We clarify that the i.i.d. property is a fundamental assumption in models derived from factor analysis (FA), including PPCA and our PGPCA. This assumption is essential for deriving the concise log-likelihood formula (2), which serves as the basis for the PGPCA EM algorithm. Nevertheless, researchers have extended static probabilistic models by incorporating temporal dependencies between the latent states $x_t$, as demonstrated in Lawrence and Hyvärinen (2005). Incorporating such temporal dependencies would fundamentally alter the modeling structure of $y_t$ and requires a complete rederivation of the PGPCA EM algorithm. Extending PGPCA to account for temporal dependence is one of our future research directions, as outlined in the Conclusion section (lines 494–498).
>
> ---
> 8. Can the assumption of normality be relaxed to allow for non-normal errors?
>
> This is another great point. A normally distributed error is a fundamental assumption in factor analysis (FA) to ensure that the probabilistic model of the observation has a closed-form expression. In our PGPCA model, the normality of the error term $r_t$ is crucial for the closed-form expression of $p(y_t | z_t)$ since the sum of two independent normal distributions is still normal. If $r_t$ is not normal, $p(y_t | z_t)$ would lack a closed-form expression, preventing us from deriving the PGPCA EM algorithm analytically and obtaining the closed-form formulas in Algorithm 1. Thus, the assumption of normal error is essential and is indeed commonly employed in many probabilistic models outside of our work.
>
> ---
> **References**
>
> Lawrence, N., & Hyvärinen, A. (2005). Probabilistic non-linear principal component analysis with Gaussian process latent variable models. Journal of Machine Learning Research 6.11.
>
> Tipping, M. E., & Bishop, C. M. (1999). Mixtures of probabilistic principal component analyzers. Neural computation, 11(2), 443-482.
>
> Park, C., Wang, M. C., & Mo, E. B. (2017). Probabilistic penalized principal component analysis. Communications for Statistical Applications and Methods, 24(2), 143-154.
>
> Nodehi, A., Golalizadeh, M., Maadooliat, M., & Agostinelli, C. (2020). Torus Probabilistic Principal Component Analysis. arXiv preprint arXiv:2008.10725.
>
> Zhang, M., & Fletcher, T. (2013). Probabilistic principal geodesic analysis. Advances in neural information processing systems, 26.

---

> > ### Comment · Reviewer_DfNu · 2024-11-22
> >
> > Thank you for your response.
> >
> > The additional information, details and experiment results, reinforce my assessment that the paper should be accepted.

---

> > > ### Author Response · Authors · 2024-11-24
> > >
> > > We again thank the reviewer for considering our manuscript and for providing insightful comments. We are glad our responses and additional results reinforced their positive assessment that our paper should be accepted.

---

### Official Review · Reviewer_CyYb · 2024-11-03

**Soundness:** 2
**Presentation:** 3
**Contribution:** 2
**Rating:** 6
**Confidence:** 4

**Summary:**

The paper's main idea is the PPCA extension containing nonlinear manifold terms. The reason for this assumption is not clear.
But the formulation and mathematical analysis are true.
Also, its performance should be seen in real data. However the experiments on real data are not clear and also they should apply it for more general data like well-known image data sets, or at least mention the class of real data that this method can work well for them in comparison with PPCA. By this version of experiments the importance and quality of the proposed method are not clear.
Also, they should report the complexity of the proposed method.

**Strengths:**

The writing is good and also the mathematics are true.

**Weaknesses:**

The paper's main idea is the PPCA extension containing nonlinear manifold terms. The reason for this assumption is not clear.
But the formulation and mathematical analysis are true.
Also, its performance should be seen in real data. However the experiments on real data are not clear and also they should apply it for more general data like well-known image data sets, or at least mention the class of real data that this method can work well for them in comparison with PPCA. By this version of experiments the importance and quality of the proposed method are not clear.
Also, they should report the complexity of the proposed method.

**Questions:**

_ experiments for real data should be done?
what is the specific class of data that this idea works good?
why they did not report the complexity?
if they report one can compare them with methods like deep factor models

**Details Of Ethics Concerns:**

its ok

---

> ### Author Response · Authors · 2024-11-22
>
> We thank the reviewer for considering our manuscript and for their feedback. We respond to the reviewer comments below:
>
> 1. The paper's main idea is the PPCA extension containing nonlinear manifold terms. The reason for this assumption is not clear. But the formulation and mathematical analysis are true.
>
> We thank the reviewer for appreciating our formulation and mathematical analysis and raising this question, which is an important point to clarify. The motivation for developing PGPCA stems from extensive evidence in neuroscience that neural population activity recorded from the brain often distributes around nonlinear manifolds (see also introduction lines 59-68). For example, neural population activity – i.e., time-series of neural data samples – has been shown to distribute around a ring manifold in the head direction system (Chaudhuri et al., 2019; Jensen et al., 2020) or around a torus manifold in the hippocampus (Gardner et al., 2022). Consequently, we hypothesized that developing a new dimensionality reduction algorithm that incorporates knowledge of a data-fitted nonlinear manifold can improve the performance of low-dimensional models in terms of describing neural/brain activity time-series data. That is why we developed PGPCA. Indeed, our data analysis (Figure 6) demonstrates that, for the same dimensionality, PGPCA models achieve significantly higher log-likelihoods than their corresponding PPCA models for real neural population activity recorded from the brains of two animals.
>
> ---
> 2. Also, its performance should be seen in real data. However the experiments on real data are not clear and also they should apply it for more general data like well-known image data sets, or at least mention the class of real data that this method can work well for them in comparison with PPCA. By this version of experiments the importance and quality of the proposed method are not clear. …Experiments for real data should be done? What is the specific class of data that this idea works good?
>
> This is an excellent point and we apologize for the lack of clarity.
>
> **Real data**: We apply PGPCA on real data recorded from the brains of two animals in the Results section 4.4. Here we applied PGPCA to the population activity of 50 and 22 neurons in the brains of two mice as they explored an open environment, respectively. The neurons were recorded from the thalamo-cortical head direction circuit of the two mice. Thus, the data consisted of the time-series of firing rates of these neurons. This dataset is publicly available (Peyrache & Buzsáki, 2015), and the preprocessing steps before applying PGPCA are primarily based on Chaudhuri et al. (2019) and explained in detail there (see also our appendix C). Our results show that PGPCA outperforms PPCA (Figure 6) by incorporating the data-fitted nonlinear manifold (in this case a loop embedded in 10-dimensional space) into the probabilistic model. Furthermore, the new geometric coordinate system we derived (i.e., PGPCA GeCOV) more accurately captures the firing rates time-series data than the Euclidean coordinate system (i.e., PGPCA EuCOV) across both mice. This result suggests that the noise not accounted for by the main loop manifold originates from the same geometric structure rather than being situated in Euclidean space. This discovery is enabled by the new feature—distribution coordinate $K(z)$—that we introduced in the PGPCA model.
>
> **Class of data for PGPCA**: As described above, since PGPCA is motivated by evidence of nonlinear manifolds underlying neural population activity (Chaudhuri et al., 2019; Jensen et al., 2020; Gardner et al., 2022), we focused on applying PGPCA to neural data from the brain in this paper. We believe PGPCA will be of broad utility in the fields of neuroscience, neuroAI, and neurotechnology. However, PGPCA is not limited to neural data time-series and can in principle be applied to any time-series dataset with a data-fitted underlying manifold. We will make this point more explicit in the Conclusion section.

---

> > ### Comment · Reviewer_CyYb · 2024-11-24
> > **about the idea and experments**
> >
> > With this explanation, it seems that this approach is applicable to specific types of data. Therefore, it is essential to specify the data scope clearly in the title. Since both the explanation and the practical implementation refer to neuronal brain data, this should be made explicit. Also, I think that using just a single practical dataset might be insufficient.

---

> ### Author Response · Authors · 2024-11-22
>
> 3. Also, they should report the complexity of the proposed method. …Why they did not report the complexity? If they report one can compare them with methods like deep factor models.
>
> We thank the reviewer for bringing up this point. Since all steps in PGPCA (Algorithm 1) are analytical, the PGPCA EM algorithm is as efficient as the standard EM for learning a linear dynamical model (linear state-space) (Roweis & Ghahramani, 1999). For example, it takes about 13 minutes on a regular desktop computer to learn a 10D PGPCA (GeCOV) model with 12000 training samples in $\mathbb{R}^{10}$ in our data analysis. While PGPCA is slower than PPCA, whose training can be done using SVD without iterations needed in EM, this trade-off is reasonable given that PGPCA incorporates a nonlinear manifold, which PPCA cannot. We will include this aspect in the paper.
>
> ---
> **References**
>
> Chaudhuri, R., Gerçek, B., Pandey, B., Peyrache, A., & Fiete, I. (2019). The intrinsic attractor manifold and population dynamics of a canonical cognitive circuit across waking and sleep. Nature neuroscience, 22(9), 1512-1520.
>
> Jensen, K., Kao, T. C., Tripodi, M., & Hennequin, G. (2020). Manifold GPLVMs for discovering non-Euclidean latent structure in neural data. Advances in Neural Information Processing Systems, 33, 22580-22592.
>
> Gardner, R. J., Hermansen, E., Pachitariu, M., Burak, Y., Baas, N. A., Dunn, B. A., ... & Moser, E. I. (2022). Toroidal topology of population activity in grid cells. Nature, 602(7895), 123-128.
>
> Peyrache, A & Buzsáki G. (2015). Extracellular recordings from multi-site silicon probes in the anterior thalamus and subicular formation of freely moving mice. CRCNS.org.
>
> Roweis, S., & Ghahramani, Z. (1999). A unifying review of linear Gaussian models. Neural computation, 11(2), 305-345.

---

> > ### Comment · Reviewer_CyYb · 2024-11-24
> > **about complexity**
> >
> > I mean the theoretical complexity. You can compare the complexity based on dimension of data and required computations mathematically and step by step

---

> > > ### Author Response · Authors · 2024-11-25
> > >
> > > 2. I mean the theoretical complexity. You can compare the complexity based on dimension of data and required computations mathematically and step by step.
> > >
> > > We apologize for misunderstanding the reviewer's question and now provide the theoretical complexity below and have added it to the manuscript (appendix C, Table 4). Assume the PGPCA model is $m$-dimensional with $M$ manifold landmarks $z_{1:M}$, and there are $T$ observations $y_{1:T}$ where $y_t \in \mathbb{R}^n$. The computational complexity of each step in one PGPCA EM iteration (Algorithm 1) is listed below.
> > >
> > > *The E-step:*
> > >
> > > |computational step|time complexity|
> > > |:--|:--|
> > > |Compute $q_i(z_j)$ for $\forall i \in [1,T]$ and $\forall j \in [1,M]$|$\mathcal{O}(T M^2)$|
> > >
> > > *The M-step:*
> > >
> > > |computational step|time complexity|
> > > |:--|:--|
> > > |Compute $\omega_j$ for $\forall j \in [1,M]$|$\mathcal{O}(T M)$|
> > > |Compute $\mathbf{\Gamma}(q)$|$\mathcal{O}(T M n^2)$|
> > > |Compute $\text{eig}(\mathbf{\Gamma}(q))$|$\mathcal{O}(n^3)$         (Parlett, 2000)|
> > > |Compute $\sigma^2$ and $\mathbf{C}$|$\mathcal{O}(n-m)$ and $\mathcal{O}(nm)$|
> > >
> > > Therefore, every PGPCA EM iteration’s computational complexity is $\mathcal{O}(T M^2) + \mathcal{O}(T M n^2)$.
> > >
> > > ---
> > > **References**
> > >
> > > Parlett, B. N. (2000). The QR algorithm. Computing in science & engineering, 2(1), 38-42.

---

> > > > ### Comment · Reviewer_CyYb · 2024-11-25
> > > > **details**
> > > >
> > > > Could you compare with PPCA and also provide details of the complexity of (14). i could not follow your answer above.
> > > > Also, its good if you write a sentence that claims the convergence theoretically

---

> ### Author Response · Authors · 2024-11-25
>
> 1. With this explanation, it seems that this approach is applicable to specific types of data. Therefore, it is essential to specify the data scope clearly in the title. Since both the explanation and the practical implementation refer to neuronal brain data, this should be made explicit. Also, I think that using just a single practical dataset might be insufficient.
>
> We thank the reviewer for this follow-up question. We agree it is critical to clarify the scope of data, and we also agree it is important to make more explicit the motivation by and application to neuronal brain data here, which we have now better done as we explain in the second point below (note our submission primary area is to the “applications to neuroscience & cognitive science” track). We clarify three points in this regard below.
>
> First, we should clarify that while our data analyses focus on neuronal brain data, our extensive simulations explore various nonlinear manifolds, distribution coordinates, and manifold latent state distributions $p(z)$ without imposing any neuron-specific restriction on the observations $y_{1:T}$. This demonstrates that PGPCA can handle a wide range of observations with data-fitted manifolds. Our mathematical derivations of PGPCA similarly do not impose any neuron-specific constraints. We chose to focus on neural data in this manuscript because that is where our inspiration/motivation for developing PGPCA came from and because we submitted our manuscript to the primary area of “applications to neuroscience & cognitive science”. However, in principle and mathematically, this does not limit the broader applicability of PGPCA in other applications, as long as these applications involve datasets that distribute around a data-fitted nonlinear manifold, because no neuron-specific criteria were imposed on the observations $y_{1:T}$ during PGPCA derivation.
>
> Second, that being said, we agree that it is important to emphasize that PGPCA is for data that are distributed around a nonlinear data-fitted geometry/manifold. In the title, this is indicated by the use of the word “geometric” and we have now also made this even more explicit in the abstract as shown in the below bold italic quotes. We also agree that we should make it more explicit that PGPCA was motivated by and applied to solving the neural distribution problem in this manuscript. We have also done that in the abstract, as seen in the bold italic quotes below:
>
> *However, in many ***neuroscience*** applications, data may be distributed around a nonlinear ***geometry (i.e., manifold)*** rather than lying in the Euclidean space around the mean. We develop Probabilistic Geometric Principal Component Analysis (PGPCA) ***for such datasets*** as a new dimensionality reduction algorithm that can explicitly incorporate knowledge about a given nonlinear manifold that is first fitted from ***these*** data.*
> …
> *These capabilities make PGPCA valuable for enhancing the efficacy of dimensionality reduction for analysis of high-dimensional data that exhibit noise and are distributed around a nonlinear manifold, ***especially for neural data***.*
>
> Please also note that the first sentence in the abstract also mentions the neuroscience focus:
>
> *Dimensionality reduction is critical across various domains of science including neuroscience.*
>
> Third, regarding real data, we should clarify that in our revisions and as suggested by reviewer DfNu, we have now significantly expanded our real-data evaluations by including 4 additional brain datasets from 4 additional different animals. Table below shows the log-likelihood values for these new datasets, similar to our original Table 2 for the original 2 datasets. We see that our conclusions again hold in these new datasets.
>
> | |Mouse17|Mouse20|Mouse24|Mouse25|
> |-|:-:|:-:|:-:|:-:|
> |GeCOV|$\mathbf{-30.407}$|$\mathbf{-19.751}$|$\mathbf{-18.687}$|$\mathbf{-20.298}$|
> |EuCOV|$-30.595$|$-19.768$|$-18.702$|$-20.356$|
> |PPCA|$-32.668$|$-21.560$|$-21.134$|$-24.977$|

---

> > ### Comment · Reviewer_CyYb · 2024-11-25
> > **About emphasizing the application area.**
> >
> > Since your claim has now been demonstrated for specific data, according to references and your statement, this must be reflected in the title. The term "geometric" is much broader than the scope of your discussion. Although you mentioned that no restrictions regarding neural data have been applied in the model, the condition of being "in the ring" is a restriction you have used. Unless you show that this restriction also holds for data beyond your specific application, I believe the title must be changed and emphasize your application.

---

> ### Author Response · Authors · 2024-11-26
>
> 1. Since your claim has now been demonstrated for specific data, according to references and your statement, this must be reflected in the title. The term "geometric" is much broader than the scope of your discussion. Although you mentioned that no restrictions regarding neural data have been applied in the model, the condition of being "in the ring" is a restriction you have used. Unless you show that this restriction also holds for data beyond your specific application, I believe the title must be changed and emphasize your application.
>
> This is an important point to address. First, the manifold $\mathbf{\phi}$ in equation (1) does not have to be a ring. For example, we have simulated data whose manifold is a Torus and PGPCA successfully captures this simulated data too (Figures 1 and 3), similar to the results with simulated data around a ring manifold. Thus, when deriving the PGPCA model and its EM learning algorithm, we do not impose a ring or any specific manifold but derive the PGPCA EM for a general manifold $\mathbf{\phi}$. In our real data, we use the ring manifold because this structure is suggested by prior literature on this dataset (Chaudhuri et al., 2019).
>
> Second, based on the reviewer’s suggestion to emphasize our application in the title, we have now updated our title to: “Probabilistic Geometric Principal Component Analysis with Application to Neural Data.” We have also updated the manuscript accordingly (please note that we cannot modify the title on this webpage, but the title on the PDF is changed already). We hope this addresses the reviewer’s comment but please let us know if you have any further comments.
>
> ---
> **References**
>
> Chaudhuri, R., Gerçek, B., Pandey, B., Peyrache, A., & Fiete, I. (2019). The intrinsic attractor manifold and population dynamics of a canonical cognitive circuit across waking and sleep. Nature neuroscience, 22(9), 1512-1520.

---

> ### Author Response · Authors · 2024-11-26
>
> 2. Could you compare with PPCA and also provide details of the complexity of (14). I could not follow your answer above. Also, its good if you write a sentence that claims the convergence theoretically.
>
> These are important points to address. First, we compute $\mathbf{\Gamma}(q)$ using equation (14):
>
> $$
> \mathbf{\Gamma}(q)
> = \frac{1}{T} \sum_{i=1}^{T} \sum_{j=1}^{M} \mathbf{\Gamma}\_{i,z_j} \times q\_i(\mathbf{z}\_j)
> \quad\text{where}\quad
> \mathbf{\Gamma}\_{i,z}
> = \mathbf{K}\_z' (\mathbf{y}\_i - \mathbf{\phi}\_z)(\mathbf{y}\_i - \mathbf{\phi}\_z)' \mathbf{K}\_z
> $$
>
> The posterior distribution $q_i(\mathbf{z}\_j)$ is already computed during the E-step. The time complexity of each $\mathbf{\Gamma}\_{i,z}$ is $\mathcal{O}(n^2)$ because $\mathbf{K}\_z \in \mathbb{R}^{n \times n}$ and $\mathbf{y}\_i, \mathbf{\phi}\_z \in \mathbb{R}^n$. Given that there are $T \times M$ different $\mathbf{\Gamma}\_{i,z}$, the overall time complexity of $\mathbf{\Gamma}(q)$ is $\mathcal{O}(T M n^2)$.
>
> Second, the computational complexity of PPCA is $\mathcal{O}(T n^2)$ because it involves computing a matrix $\mathbf{S}$ (Tipping & Bishop, 1999), which is analogous to our $\mathbf{\Gamma}(q)$, as described in the derivation of the PGPCA M-step (section 3.5). However, PPCA can be solved without iterations (unlike EM) by performing PCA on $\mathbf{S}$, making it more computationally efficient compared to PGPCA. This is reasonable and expected since our PGPCA provides the capability to incorporate arbitrary data-fitted manifolds, which inherently increases both the modeling and computational complexity compared to PPCA. We have now provided the computational complexity of PPCA in appendix C where we discuss the computational complexity of PGPCA.
>
> Third, the convergence of our PGPCA EM is theoretically guaranteed by the properties of the EM algorithm. The evidence lower bound (ELBO) $\mathcal{L}^E$ in equation (5) is bounded from above by the log-likelihood $\mathcal{L} = \ln p(y_{1:T})$, which is finite. Additionally, because our E-step can find the posterior distribution $q_i(\mathbf{z}_j)$ and our M-step can analytically find the parameters that maximize the ELBO, together our E-step and M-step ensure a monotonic increase in ELBO $\mathcal{L}^E$ with each iteration. Consequently, the sequence of $\mathcal{L}^E$ values is monotonically increasing and bounded from above, ensuring PGPCA EM convergence by the completeness property of real numbers (Rudin, 1964). We have now mentioned this in appendix C where we discuss the computational complexity of PGPCA.
>
> ---
> **References**
>
> Tipping, M. E., & Bishop, C. M. (1999). Probabilistic principal component analysis. Journal of the Royal Statistical Society Series B: Statistical Methodology, 61(3), 611-622.
>
> Rudin, W. (1964). Principles of mathematical analysis (Vol. 3). New York: McGraw-hill.

---

### Official Review · Reviewer_VSUw · 2024-11-04

**Soundness:** 3
**Presentation:** 3
**Contribution:** 3
**Rating:** 8
**Confidence:** 3

**Summary:**

In this work, the author proposes a Probabilistic Geometric Principal Component Analysis (PGPCA) method which can be seen as an extension of PPCA with a better description of data distributions by incorporating a nonlinear manifold geometry. A data-driven EM algorithm is also proposed to solve the PGPCA problem. Experimental results verify the performance of the PGPCA is better than that of PPCA.

**Strengths:**

The method and algorithms are technically sound, with reasonable insights and solutions.

**Weaknesses:**

Comparison with other nonlinear PPCA methods is not provided.

**Questions:**

1. As some works focus on nonlinear PPCA, such as

Lawrence, Neil, and Aapo Hyvärinen. "Probabilistic non-linear principal component analysis with Gaussian process latent variable models." Journal of Machine Learning Research 6.11 (2005).

Zhang, Jingxin, et al. "An improved mixture of probabilistic PCA for nonlinear data-driven process monitoring." IEEE transactions on cybernetics 49.1 (2017): 198-210.

how is the performance of the proposed PGPCA compared with these nonlinear PPCA methods?


2. How does PGPCA compare with other methods in terms of computational efficiency?

---

> ### Author Response · Authors · 2024-11-22
>
> We thank the reviewer for considering our manuscript and for their feedback. We respond to the reviewer comments below:
>
> 1. Comparison with other nonlinear PPCA methods is not provided. As some works focus on nonlinear PPCA, such as
>
> * Lawrence, Neil, and Aapo Hyvärinen. "Probabilistic non-linear principal component analysis with Gaussian process latent variable models." Journal of Machine Learning Research 6.11 (2005).
>
> * Zhang, Jingxin, et al. "An improved mixture of probabilistic PCA for nonlinear data-driven process monitoring." IEEE transactions on cybernetics 49.1 (2017): 198-210.
>
>    How is the performance of the proposed PGPCA compared with these nonlinear PPCA methods?
>
> We thank the reviewer for suggesting these two references. Zhang et al. (2017) is focused on using an existing method, mixture PPCA, to integrate two monitoring statistics in order to address a fault diagnosis problem. As such, this work does not develop a new probabilistic model. This mixture PPCA (Tipping & Bishop, 1999a) has already been cited in our paper during the derivation of the M-step of the PGPCA EM algorithm in Section 3.5. Therefore, its connection to our PGPCA has been addressed. We will also add Zhang et al. (2017) there to show an application of mixture PPCA.
>
> In contrast, Lawrence (2005) does indeed introduce a new algorithm, the Gaussian Process Latent Variable Model (GP-LVM), inspired by PPCA. We have now added a comparison with GP-LVM by applying it to our mouse neural data, showing our PGPCA outperforms it for modeling of data distribution. For GP-LVM, we initialized it by either PCA or Isomap as suggested in Lawrence (2005). We found that the log-likelihood achieved by GP-LVM was consistently much smaller (less than $-200$) compared to that of our PGPCA (greater than $-35$). This is because for an $n$-dimensional observation $y_t$, GP-LVM models the time series of each observation dimension $y_{1:T}(d) \phantom{s} (d = 1 \dots n)$ separately and in $\mathbb{R}^T$, rather than modeling the distribution of $y_t$ in the embedding space $\mathbb{R}^n$. As a result, it does not efficiently capture the manifold underlying the observations $y_{1:T}$ as our PGPCA does. Note that this is expected, as the goal of GP-LVM is to find a nonlinear low-dimensional description of data for classification, rather than to describe the data distribution in its own embedding space. Indeed, as shown in Lawrence (2005), the log-likelihood of a trained GP-LVM can be very negative around $-500$. As such, our PGPCA outperforms GP-LVM for modeling of data distribution, and thus provides a tool with a distinct goal compared with GP-LVM.
>
> Furthermore, GP-LVM's training time poses a significant bottleneck due to its non-convex cost function and the large number of latent states requiring optimization. For instance, on a regular desktop computer, training a 4D GP-LVM model (i.e., with a latent state dimension of 4) with 3000 time-samples ($T=3000$) takes 532 minutes (approximately 8.9 hours) to converge, as it involves fitting $4 \times 3000 = 12000$ latent state elements. This is because GP-LVM treats its latent states as parameters in the optimization process. In contrast, training our 10D PGPCA (GeCOV) model with 12000 samples takes only about 13 minutes to converge. That is, with more training samples, our PGPCA still converges much faster than GP-LVM. This highlights that PGPCA is better suited for modeling data distributions by leveraging data-fitted manifold information. Indeed, GP-LVM is typically used for categorization tasks rather than for distribution modeling.
>
> Nonetheless, GP-LVM is a nonlinear probabilistic model inspired by PPCA, and we will include it in the Related Work section. We will also discuss the distinct goal of GP-LVM there.
>
> ---
> 2. How does PGPCA compare with other methods in terms of computational efficiency?
>
> We thank the reviewer for bringing up this point. Since all steps in PGPCA (Algorithm 1) are analytical, the PGPCA EM algorithm is as efficient as the standard EM for learning a linear dynamical model (linear state-space) (Roweis & Ghahramani, 1999). For example, it takes about 13 minutes on a regular desktop computer to learn a 10D PGPCA (GeCOV) model with 12000 training samples in $\mathbb{R}^{10}$ in our data analysis. While PGPCA is slower than PPCA, whose training can be done using SVD without iterations, this trade-off is reasonable given that PGPCA incorporates a nonlinear manifold, which PPCA does not. We will include this aspect in the paper.
>
> ---
> **References**
>
> Tipping, M. E., & Bishop, C. M. (1999). Mixtures of probabilistic principal component analyzers. Neural computation, 11(2), 443-482.
>
> Roweis, S., & Ghahramani, Z. (1999). A unifying review of linear Gaussian models. Neural computation, 11(2), 305-345.

---

> > ### Comment · Reviewer_VSUw · 2024-11-24
> >
> > Thank the authors for their responses. The authors have addressed my concern and I have raised my score. I have no further questions.

---

> > > ### Author Response · Authors · 2024-11-24
> > >
> > > We are glad our responses addressed the reviewer’s concerns. We thank the reviewer again for their insightful feedback and for raising their score.

---

### Meta-Review · Area_Chair_SeEu · 2024-12-23

**Metareview:**

The  paper proposes Probabilistic Geometric Principal Component Analysis (PGPCA), which is a generalization of Probabilistic PCA (PPCA) to handle the case the data is distributed around a nonlinear manifold. A data-driven EM algorithm is proposed to solve the PGPCA problem. Experimental results are provided and verify that the performance of the PGPCA is better than that of PPCA.

Note: after the discussion period, the authors changed the title of their paper (PDF file) to "Probabilistic Geometric Principal Component Analysis with Application to Neural Data"

Reviewers generally agree that the proposed extension PGPCA of PPCA is novel and well- explained, the mathematical derivation of the EM-algorithm is concise and understandable, and the experimental results appear promising.

I do have reservations concerning the paper:

- The scope of the proposed method is limited to data of a specific type: distributed around a fixed nonlinear manifold. The method itself is still a linear method.

- The derivation of the EM-algorithm is a straightforward extension of that in PPCA.

- The experimental results on real data are very limited, especially given the fact that the paper is aimed towards "Application to Neural Data"

However, the method could be of interest to the neuroscience community.

**Additional Comments On Reviewer Discussion:**

- Reviewers VSUw,  DfNu: Comparison with other nonlinear PPCA methods is not provided (only comparison with PPCA is given). In response, the authors applied Gaussian Process Latent Variable Model (GP-LVM) (Lawrence, 2005), which gives inferior results (however the aim of GP-LVM is different, namely to find a nonlinear low-dimensional description of data for classification)

- Reviewer CyYb: This approach is applicable to specific types of data. The authors confirm that  PGPCA is for data that are distributed around a nonlinear data-fitted geometry/manifold and consequently updated the title of the paper to “Probabilistic Geometric Principal Component Analysis with Application to Neural Data"

 - Reviwer CyYb, DfNu: a single practical dataset might be insufficient. The authors added experimental results on 4 additional mice (the original results reported 2 mice)

---

### Decision · Program_Chairs · 2025-01-22

Accept (Spotlight)